# Relative Flatness and Generalization

**Henning Petzka**[*]
Lund University, Sweden
`henning.petzka@math.lth.se`

**Michael Kamp**[*]
CISPA Helmholtz Center for Information Security,
Germany and Monash University, Australia
`michael.kamp@monash.edu`

**Linara Adilova**
Ruhr University Bochum, Germany
and Fraunhofer IAIS

**Cristian Sminchisescu**
Lund University, Sweden
and Google Research, Switzerland

**Mario Boley**
Monash University, Australia

## Abstract

Flatness of the loss curve is conjectured to be connected to the generalization ability of machine learning models, in particular neural networks. While it has been empirically observed that flatness measures consistently correlate strongly with generalization, it is still an open theoretical problem why and under which circumstances flatness is connected to generalization, in particular in light of reparameterizations that change certain flatness measures but leave generalization unchanged. We investigate the connection between flatness and generalization by relating it to the interpolation from representative data, deriving notions of representativeness, and feature robustness. The notions allow us to rigorously connect flatness and generalization and to identify conditions under which the connection holds. Moreover, they give rise to a novel, but natural relative flatness measure that correlates strongly with generalization, simplifies to ridge regression for ordinary least squares, and solves the reparameterization issue.

## 1 Introduction

Flatness of the loss curve has been identified as a potential predictor for the generalization abilities of machine learning models [6, 10, 11]. In particular for neural networks, it has been repeatedly observed that generalization performance correlates with measures of flatness, i.e., measures that quantify the change in loss under perturbations of the model parameters [4, 8, 16, 21, 34, 39, 41, 44]. In fact, Jiang et al. [14] perform a large-scale empirical study and find that flatness-based measures have a higher correlation with generalization than alternatives like weight norms, margin-, and optimization-based measures. It is an open problem why and under which circumstances this correlation holds, in particular in the light of negative results on reparametrizations of ReLU neural networks [5]: these reparameterizations change traditional measures of flatness, yet leave the model function and its generalization unchanged, making these measures unreliable. We present a novel and rigorous approach to understanding the connection between flatness and generalization by relating it to the interpolation from representative samples. Using this theory we, for the first time, identify conditions under which flatness explains generalization. At the same time, we derive a measure of *relative flatness* that simplifies to ridge/Tikhonov regularization for ordinary least squares [36], and resolves

---

[*]equal contribution

35th Conference on Neural Information Processing Systems (NeurIPS 2021).

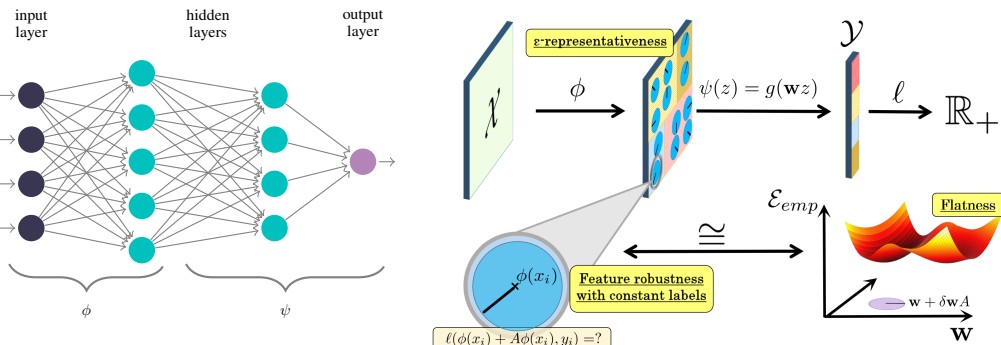

Figure 1: Decomposition of $f = \psi \circ \phi$ into a feature extractor $\phi$ and a model $\psi$ for neural networks.

Figure 2: Overview: We theoretically connect a notion of representative data with a notion of feature robustness and a novel measure of flatness of the loss surface.

the reparametrization issue for ReLU networks [5] by appropriately taking the norm of parameters into account as suggested by Neyshabur et al. [25].

Formally, we connect flatness of the loss surface to the *generalization gap* $\mathcal{E}_{gen}(f, S) = \mathcal{E}(f) - \mathcal{E}_{emp}(f, S)$ of a *model* $f : \mathcal{X} \to \mathcal{Y}$ from a *model class* $\mathcal{H}$ with respect to a twice differentiable *loss function* $\ell : \mathcal{Y} \times \mathcal{Y} \to \mathbb{R}_+$ and a finite sample set $S \subseteq \mathcal{X} \times \mathcal{Y}$, where

$$\mathcal{E}(f) = \mathbb{E}_{(x,y) \sim \mathcal{D}} \left[ \ell(f(x), y) \right] \ \text{ and } \ \mathcal{E}_{emp}(f, S) = \frac{1}{|S|} \sum_{(x,y) \in S} \ell(f(x), y) \ .$$

That is, $\mathcal{E}_{gen}(f, S)$ is the difference between the *risk* $\mathcal{E}(f)$ and the *empirical risk* $\mathcal{E}_{emp}(f, S)$ of $f$ on a finite sample set $S$ drawn iid. according to a *data distribution* $\mathcal{D}$ on $\mathcal{X} \times \mathcal{Y}$. To connect flatness to generalization, we start by decomposing the generalisation gap into two terms, a *representativeness* term that quantifies how well a distribution $\mathcal{D}$ can be approximated using distributions with local support around sample points and a *feature robustness* term describing how small changes of feature values affect the model's loss. Here, feature value refers to the implicitly represented features by the model, i.e., we consider models that can be expressed as $f(x) = \psi(\mathbf{w}, \phi(x)) = g(\mathbf{w}\phi(x))$ with a feature extractor $\phi$ and a model $\psi$ (which includes linear and kernel models, as well as most neural networks, see Fig. 1). With this decomposition, we measure the generalization ability of a particular model by how well its interpolation between samples in feature space fits the underlying data distribution. We then connect feature robustness (a property of the feature space) to flatness (a property of the parameter space) using the following key identity: Multiplicative perturbations in feature space by arbitrary matrices $A \in \mathbb{R}^{m \times m}$ correspond to perturbations in parameter space, i.e.,

$$\psi(\mathbf{w}, \phi(x) + A\phi(x)) = g(\mathbf{w}(\phi(x) + A\phi(x))) = g((\mathbf{w} + \mathbf{w}A)\phi(x)) = \psi(\mathbf{w} + \mathbf{w}A, \phi(x)) \ . \quad (1)$$

Using this key equation, we show that feature robustness is approximated by a novel, but natural, loss Hessian-based *relative flatness* measure under the assumption that the distribution can be approximated by locally constant labels. Under this assumption and if the data is representative, then flatness is the main predictor of generalization (see Fig. 2 for an illustration).

This offers an explanation for the correlation of flatness with generalization on many real-world data distributions for image classification [14, 21, 26], where the assumption of locally constant labels is reasonable (the definition of adversarial examples [35] even hinges on this assumption). This dependence on locally constant labels has not been uncovered by previous theoretical analysis [26, 37]. Moreover, we show that the resulting relative flatness measure is invariant to linear reparameterization and has a stronger correlation with generalization than other flatness measures [14, 21, 26]. Other measures have been proposed that similarly achieve invariance under reparameterizations [21, 37], but the Fisher-Rao norm [21] is lacking a strong theoretical connection to generalization, our measure sustains a more natural form than normalized sharpness [37] and for neural networks, it considers only a single layer, given by the decomposition of $f$ (including the possibility of choosing the input layer when $\phi = id_{\mathcal{X}}$). An extended comparison to related work is provided in Appdx. A.

The limitations of our analysis are as follows. We assume a noise-free setting where for each $x \in \mathcal{X}$ there is a unique $y = y(x) \in \mathcal{Y}$ such that $P_{x,y \sim D}(y|x) = 1$, and this assumption is also extended to

the feature space of the given model, i.e., we assume that $\phi(x) = \phi(x')$ implies $y(x) = y(x')$ for all $x, x' \in \mathcal{X}$ and write $y(x) = y(\phi(x))$. Moreover, we assume that the marginal distribution $\mathcal{D}_{\mathcal{X}}$ is described by a density function $p_{\mathcal{D}}(x)$, that $f(x) = \psi(\mathbf{w}, \phi(x)) = g(\mathbf{w}\phi(x))$ is a local minimizer of the empirical risk on $S$, and that $g, \psi, \phi$ are twice differential. Quantifying the representativeness of a dataset precisely is challenging since the data distribution is unknown. Using results from density estimation, we derive a worst-case bound on representativeness for all data distributions that fulfill mild regularity assumptions in feature space $\phi(\mathcal{X})$, i.e., a smooth density function $p_{\phi(\mathcal{D})}$ such that $\int_{z \in \phi(\mathcal{X})} \left| \nabla^2 \left( p_{\phi(\mathcal{D})}(z) ||z||^2 \right) \right| dz$ and $\int_{z \in \phi(\mathcal{X})} p_{\phi(\mathcal{D})}(z)/||z||^m \, dz$ are well-defined and finite. This yields a generalization bound incorporating flatness. In contrast to the common bounds of statistical learning theory, the bound depends on the feature dimension. The dimension-dependence is a result of the interpolation approach (applying density estimation uniformly over all distributions that satisfy the mild regularity assumptions). The bound is consistent with the no-free-lunch theorem and the convergence rate derived by Belkin et al. [2] for a model based on interpolations. In practical settings, representativeness can be expected to be much smaller than the worst-case bound, which we demonstrate by a synthetic example in Sec. 6. Generally, it is a bound that remains meaningful in the interpolation regime [1, 3, 24], where traditional measures of generalization based on the empirical risk and model class complexity are uninformative [22, 42].

**Contribution.** In summary, this paper rigorously connects flatness of the loss surface to generalization and shows that this connection requires feature representations such that labels are (approximately) locally constant, which is also validated in a synthetic experiment (Sec. 6). The empirical evaluation shows that this flatness and an approximation to representativeness can tightly bound the generalization gap. Our contributions are: (i) the rigorous connection of flatness and generalization; (ii) novel notions of representativeness and feature robustness that capture the extent to which a model's interpolation between samples fits the data distribution; and (iii) a novel flatness measure that is layer- and neuron-wise reparameterization invariant, reduces to ridge regression for ordinary least squares, and outperforms state-of-the-art flatness measures on CIFAR10.

## 2   Representativeness

In this section, we formalize when a sample set $S$ is representative for a data distribution $\mathcal{D}$.

**Partitioning the input space.** We choose a partition $\{V_i \,|\, i = 1, \ldots, |S|\}$ of $\mathcal{X}$ such that each element of this partition $V_i$ contains exactly one of the samples $x_i$ from $S$. The distribution can then be described by a set of densities $p_i(x) = \frac{1}{\alpha_i} \cdot p_{\mathcal{D}}(x) \cdot \mathbf{1}_{V_i}(x)$ with support contained in $V_i$ (where $\mathbf{1}_{V_i}(x) = 1$ if $x \in V_i$ and 0 otherwise) and with normalizing factor $\alpha_i = \int_{V_i} p_{\mathcal{D}}(x)dx$. Then the risk decomposes as $\mathcal{E}(f) = \sum_{i=1}^{|S|} \alpha_i \cdot \mathbb{E}_{x \sim p_i}[\ell(f(x), y(x))]$. Since $x_i \in V_i$ for each $i$, we can change variables and consider density functions $\lambda_i^*(\xi) = p_i(x_i + \xi)$ with support in a neighborhood around the origin of $\mathcal{X}$. The risk then decomposes as

$$\mathcal{E}(f) = \sum_{i=1}^{|S|} \alpha_i \cdot \mathbb{E}_{\xi \sim \lambda_i^*}[\ell(f(x_i + \xi), y(x_i + \xi))] \ . \tag{2}$$

Starting from this identity, we formalize an approximation to the risk: In a practical setting, the distribution $p_{\mathcal{D}}$ is unknown and hence, in the decomposition (2), we have unknown densities $\lambda_i^*$ and unknown normalization factors $\alpha_i$. We assume that each neighborhood contributes equally to the loss, i.e., we approximate each $\alpha_i$ with $\frac{1}{|S|}$. Then, given a sample set $S$ and an $|S|$-tuple $\Lambda = (\lambda_i)_{1 \leq i \leq |S|}$ of "local" probability density functions on $\mathcal{X}$ with support $supp(\lambda_i)$ in a neighborhood around the origin $0_{\mathcal{X}}$, we call the pair $(S, \Lambda)$ $\epsilon$-*representative for* $\mathcal{D}$ with respect to a model $f$ and loss $\ell$ if $|\mathcal{E}_{Rep}(f, S, \Lambda)| \leq \epsilon$, where

$$\mathcal{E}_{Rep}(f, S, \Lambda) = \mathcal{E}(f) - \sum_{i=1}^{|S|} \frac{1}{|S|} \cdot \mathbb{E}_{\xi \sim \lambda_i} [\ell(f(x_i + \xi), y(x_i + \xi))] \ . \tag{3}$$

If the partitions $V_i$ and the distributions $\lambda_i$ are all chosen optimal so that the approximation $\alpha_i = \frac{1}{|S|}$ is exact and $\lambda_i = \lambda_i^*$, then $\mathcal{E}_{Rep}(f, S, \Lambda) = 0$ by (2). If the support of each $\lambda_i$ is decreased to the origin so that $\lambda_i = \delta_0$ is a Dirac delta function, then $\mathcal{E}_{Rep}(f, S, \Lambda) = \mathcal{E}_{gen}(f, S)$ equals the

generalization gap. For density functions with an intermediate support, the generalization gap can be decomposed into representativeness and the expected deviation of the loss around the sample points:

$$\mathcal{E}_{gen}(f,S) = \mathcal{E}_{Rep}(f,S,\Lambda) + \sum_{i=1}^{|S|} \frac{1}{|S|} \cdot \mathbb{E}_{\xi \sim \lambda_i}[\ell(f(x_i+\xi), y(x_i+\xi)) - \ell(f(x_i), y_i)]$$

The main idea of our approach to understand generalization is to use this equality and to control both representativeness and expected loss deviations for a suitable $|S|$-tuple of distributions $\Lambda$.

**From input to feature space.** An interesting aspect of $\epsilon$-representativeness is that it can be considered in a feature space instead of the input space. For a model $f = (\psi \circ \phi) : \mathcal{X} \to \mathcal{Y}$, we can apply our notion to the feature space $\phi(\mathcal{X})$ (see Fig. 1 for an illustration). This leads to the notion of $\epsilon$-representativeness in feature space defined for an $|S|$-tuple $\Lambda^\phi = (\lambda_i^\phi)_{1 \leq i \leq |S|}$ of densities on $\phi(\mathcal{X})$ by replacing $x_i$ with $\phi(x_i)$ in (3), which we denote by $\mathcal{E}_{Rep}^\phi(f,S,\Lambda^\phi)$. By measuring representativeness in a feature space, this becomes a notion of both data and feature representation. In particular, it assumes that a target output function $y(\phi(x))$ also exists for the feature space. We can then decompose the generalization gap $\mathcal{E}_{gen}(f)$ of $f = (\psi \circ \phi)$ into

$$\mathcal{E}_{Rep}^\phi(f,S,\Lambda^\phi) + \left( \frac{1}{|S|} \sum_{i=1}^{|S|} \mathbb{E}_{\xi \sim \lambda_i^\phi} \left[ \ell(\psi(\phi(x_i)+\xi), y(\phi(x_i)+\xi)) - \ell(f(x_i), y_i) \right] \right)$$

The second term is determined by how the loss changes under small perturbations in the feature space for the samples in $S$. As before, for $\lambda_i = \delta_0$ the term in the bracket vanishes and $\mathcal{E}_{Rep}^\phi(f,S,\Lambda^\phi) = \mathcal{E}_{gen}$. But the decomposition becomes more interesting for distributions with support of nonzero measure around the origin. If the true distribution can be interpolated efficiently in feature space from the samples in $S$ with suitable $\lambda_i^\phi$ so that $\mathcal{E}_{Rep}^\phi(f,S,\Lambda^\phi) \approx 0$, then the term in the bracket approximately equals the generalization gap and the generalization gap can be estimated from local properties in feature space around sample points.

## 3 Feature Robustness

Having decomposed the generalisation gap into a representativness and a second term of loss deviation, we now develop a novel notion of feature robustness that is able to bound the second term for specific families of distributions $\Lambda$ using key equation (1). Our definition of *feature robustness* for a model $f = (\psi \circ \phi) : \mathcal{X} \to \mathcal{Y}$ depends on a small number $\delta > 0$, a sample set $S$ and a *feature selection* defined by a matrix $A \in \mathbb{R}^{m \times m}$ of *operator norm* $||A|| \leq 1$. With feature perturbations $\phi_A(x) = (I + A)\phi(x)$ and

$$\mathcal{E}_{\mathcal{F}}^\phi(f,S,A) := \frac{1}{|S|} \sum_{i=1}^{|S|} \left[ \ell(\psi(\phi_A(x_i)), y[\phi_A(x_i)]) - \ell(f(x_i), y_i) \right], \tag{4}$$

the definition of feature robustness is given as follows.

**Definition 1.** *Let $\ell : \mathcal{Y} \times \mathcal{Y} \to \mathbb{R}_+$ denote a loss function, $\epsilon$ and $\delta$ two positive (small) real numbers, $S \subseteq \mathcal{X} \times \mathcal{Y}$ a finite sample set, and $A \in \mathbb{R}^{m \times m}$ a matrix. A model $f(x) = (\psi \circ \phi)(x)$ with $\phi(\mathcal{X}) \subseteq \mathbb{R}^m$ is called $((\boldsymbol{\delta}, \boldsymbol{S}, \boldsymbol{A}), \boldsymbol{\epsilon})$-feature robust, if $\left| \mathcal{E}_{\mathcal{F}}^\phi(f,S,\alpha A) \right| \leq \epsilon$ for all $0 \leq \alpha \leq \delta$. More generally, for a probability distribution $\mathcal{A}$ on perturbation matrices in $\mathbb{R}^m$, we define*

$$\mathcal{E}_{\mathcal{F}}^\phi(f,S,\mathcal{A}) = \mathbb{E}_{A \sim \mathcal{A}} \left[ \mathcal{E}_{\mathcal{F}}^\phi(f,S,A) \right] ,$$

*and call the model $((\boldsymbol{\delta}, \boldsymbol{S}, \boldsymbol{\mathcal{A}}), \boldsymbol{\epsilon})$-feature robust on average over $\mathcal{A}$, if $\left| \mathcal{E}_{\mathcal{F}}^\phi(f,S,\alpha \mathcal{A}) \right| \leq \epsilon$ for $0 \leq \alpha \leq \delta$.*

Given a feature extractor $\phi$, feature robustness measures the performance of $\psi$ when feature values are perturbed (with constant feature extractor $\phi$). This local robustness at sample points differs from the robustness of Xu and Mannor [40] that requires a data-independent partitioning of the input space. The matrix $A$ in feature robustness determines which feature values shall be perturbed. For

each sample, the perturbation is linear in the expression of the feature. Thereby, we only perturb features that are relevant for the output for a given sample and leave feature values unchanged that are not expressed. For $\phi$ mapping into an intermediate layer of a neural network, traditionally, the activation values of a neuron are considered as feature values, which corresponds to a choice of $A$ as a projection matrix. However, it was shown by Szegedy et al. [35] that, for any other direction $v \in \mathbb{R}^m, ||v|| = 1$, the values $\langle \phi(x), v \rangle$ obtained from the projection $\phi(x)$ onto $v$, can be likewise semantically interpreted as a feature. This motivates the consideration of general *feature matrices $A$*.

**Distributions on feature matrices induce distributions on the feature space**    Feature robustness is defined in terms of feature matrices (suitable for an application of (1) to connect perturbations of features with perturbations of weights), while the approach exploiting representative data from Section 2 considers distributions on feature vectors, cf. (3). To connect feature robustness to the notion of $\epsilon$-representativeness, we specify for any distribution $\mathcal{A}$ on matrices $A \in \mathbb{R}^{m \times m}$ an $|S|$-tuple $\Lambda_{\mathcal{A}} = (\lambda_i)$ of probability density functions $\lambda_i$ on the feature space $\mathbb{R}^m$ with support containing the origin. Multiplication of a feature matrix with a feature vector $\phi(x_i)$ defines a feature selection $A\phi(x_i)$, and for each $z \in \mathbb{R}^m$ there is some feature matrix $A$ with $\phi(x_i) + z = \phi(x_i) + A\phi(x_i)$ (unless $\phi(x_i) = 0$). Our choice for distributions $\lambda_i$ on $\mathbb{R}^m$ are therefore distributions that are induced via multiplication of feature vectors $\phi(x_i) \in \mathbb{R}^m$ with matrices $A \in \mathbb{R}^{m \times m}$ sampled from a distribution on feature matrices $\mathcal{A}$ . Formally, we assume that a Borel measure $\mu_A$ is defined by a probability distribution $\mathcal{A}$ on matrices $\mathbb{R}^{m \times m}$. We then define Borel measures $\mu_i$ on $\mathbb{R}^m$ by $\mu_i(C) = \mu_A(\{A \mid A\phi(x_i) \in C\})$ for Borel sets $C \subseteq \mathbb{R}^m$. Then $\lambda_i$ is the probability density function defined by the Borel measure $\mu_i$. As a result, we have for each $i$ that

$$\mathbb{E}_{A \sim \mathcal{A}} \left[ \ell(\psi(\phi_A(x_i)), y(\phi_A(x_i))) \right] = \mathbb{E}_{z \sim \lambda_i} \left[ \ell(\psi(\phi(x_i) + z), y(\phi(x_i) + z)) \right]$$

**Feature robustness and generalization.**    With this construction and a distribution $\Lambda_{\mathcal{A}}$ on the feature space induced by a distribution $\mathcal{A}$ on feature matrices, we have that

$$\mathcal{E}(f) = \mathcal{E}_{emp}(f, S) + \mathcal{E}_{Rep}^{\phi}(f, S, \Lambda_{\mathcal{A}}) + \mathcal{E}_{\mathcal{F}}^{\phi}(f, S, \mathcal{A}) \tag{5}$$

Here, $\mathcal{A}$ can be any distribution on feature matrices, which can be chosen suitably to control how well the corresponding mixture of local distributions approximates the true distribution. The third term then measures how robust the model is in expectation over feature changes for $A \sim \mathcal{A}$. In particular, if $\mathcal{E}_{Rep}^{\phi}(f, S, \Lambda_{\mathcal{A}}) \approx 0$, then $\mathcal{E}_{gen}(f, S) \approx \mathcal{E}_{\mathcal{F}}^{\phi}(f, S, \mathcal{A})$ and the generalization gap is determined by feature robustness. We end this section by illustrating how distributions on feature matrices induce natural distributions on the feature space. The example will serve in Sec. 5 to deduce a bound on $\mathcal{E}_{Rep}^{\phi}(f, S, \Lambda_{\mathcal{A}})$ from kernel density estimation.

**Example: Truncated isotropic normal distributions** are induced by a suitable distribution on feature matrices. We consider probability distributions $\mathcal{K}_{\delta||\phi(x_i)||}$ on feature vectors $z \in \mathbb{R}^m$ in the feature space defined by densities $k_{\delta||\phi(x_i)||}(0, z)$ with smooth rotation-invariant kernels, bounded support and bandwidth $h$:

$$k_h(z_i, z) = \frac{1}{h^m} \cdot k\left(\frac{||z_i - z||}{h}\right) \cdot \mathbb{1}_{||z_i - z|| < h} \tag{6}$$

with $\mathbb{1}_{||z_i - z|| < h} = 1$ when $||z - z_i|| < h$ and 0 otherwise, and such that $\int_{z \in \mathbb{R}^m} k_h(z_0, z) \, dz = 1$ for all $z_0$. An example for such a kernel is a truncated isotropic normal distribution with variance $h^2 \sigma^2 I$, $k_h(z_i, z) = \mathcal{N}(z_i, h^2 \sigma^2)(z)$. The following result states that the densities in (6) can indeed be induced by distributions on feature matrices, which will enable us to connect feature robustness with $\epsilon$-representativeness.

**Proposition 2.** *Let $S^{\phi} = \{\phi(x_i) \mid x_i \in S\}$ be a set of feature vectors in $\mathbb{R}^m$. With $k_h$ defined as in (6), let $\lambda_i(z) = k_{\delta||\phi(x_i)||}(0, z)$ define an $|S|$-tuple $\Lambda_{\delta}$ of densities. Then there exists a distribution $\mathcal{A}_{\delta}$ on matrices in $\mathbb{R}^{m \times m}$ of norm less than $\delta$ such that for each $i = 1, \ldots, |S|$,*

$$\mathbb{E}_{A \sim \mathcal{A}_{\delta}} \left[ \ell(\psi(\phi_A(x_i)), y(\phi_A(x_i))) \right] = \mathbb{E}_{\xi \sim \lambda_i} \left[ \ell(\psi(\phi(x_i) + \xi), y(\phi(x_i) + \xi)) \right]$$

The technical proof is deferred to the appendix, but we describe the distribution $\mathcal{A}_{\delta}$ on matrices for later use: The desired distribution is defined on the set of matrices of the form $rO$ for a real number $r$

and an orthogonal matrix $O$ (i.e. $OO^T = O^T O = I$) as a product measure combining the (unique) Haar measure on the set of orthogonal matrices $\mathcal{O}(m)$ with a suitable distribution on $\mathbb{R}$. The Haar measure on $\mathcal{O}(m)$ induces the uniform measure on a sphere of radius $r$ via multiplication with a vector of length $r$ [17], and we choose a measure on $\mathbb{R}$ to match the radial change of the kernel $k_h$.

## 4 Relative Flatness of the Loss Surface

Flatness is a property of the parameter space quantifying the change in loss under small parameter perturbations, classically measured by the trace of the loss Hessian $Tr(H)$, where $H$ is the matrix containing the partial second derivatives of the empirical risk with respect to all parameters of the model. In order to connect feature robustness (a property of the feature space) to flatness, we present how key equation (1) translates to the empirical risk: For a model $f(x, \mathbf{w}) = \psi(\mathbf{w}, \phi(x)) = g(\mathbf{w}\phi(x))$ with parameters $\mathbf{w} \in \mathbb{R}^{d \times m}$ and $g : \mathbb{R}^d \to \mathcal{Y}$ a function on a matrix product of parameters $\mathbf{w}$ and a feature representation $\phi : \mathcal{X} \to \mathbb{R}^m$ and any feature matrix $A \in \mathbb{R}^{m \times m}$ we have that

$$
\mathcal{E}_{emp}(\mathbf{w}+\mathbf{w}A, \phi(S)) = \frac{1}{|S|} \sum_{i=1}^{|S|} \ell(\psi(\mathbf{w} + \mathbf{w}A, \phi(x_i)), y_i)
$$

$$
= \frac{1}{|S|} \sum_{i=1}^{|S|} \ell(\psi(\mathbf{w}, \phi(x_i) + A\phi(x_i)), y_i) = \frac{1}{|S|} \sum_{i=1}^{|S|} \ell(\psi(\mathbf{w}, \phi_A(x_i)), y_i)
$$

(7)

Subtracting $\mathcal{E}_{emp}(\mathbf{w}, \phi(S)) = \frac{1}{|S|} \sum_{i=1}^{|S|} \ell(\psi(\mathbf{w}, \phi(x_i)), y_i)$, we can recognize feature robustness (4) on the right side of this equality when labels are constant under perturbations of the features, i.e. $y(\phi_A(x_i)) = y_i$. In other words, flatness $\mathcal{E}_{emp}(\mathbf{w} + \mathbf{v}, \phi(S)) - \mathcal{E}_{emp}(\mathbf{w}, \phi(S))$ describes the performance of a model function on perturbed feature vectors while holding labels constant. We proceed to introduce a novel, but natural, loss Hessian-based flatness measure that approximates feature robustness, given that the underlying data distribution $\mathcal{D}$ satisfies the assumption of locally constant labels.

With $\mathbf{w}_s = (w_{s,t})_t \in \mathbb{R}^{1 \times m}$ denoting the $s$-th row of the parameter matrix $\mathbf{w}$, we let $H_{s,s'}(\mathbf{w}, \phi(S)) \in \mathbb{R}^{m \times m}$ denote the Hessian matrix containing all partial second derivatives of the empirical risk $\mathcal{E}_{emp}(\mathbf{w}, \phi(S))$ with respect to weights in rows $\mathbf{w}_s$ and $\mathbf{w}_{s'}$, i.e.

$$
H_{s,s'}(\mathbf{w}, \phi(S)) = \left[ \frac{\partial^2 \mathcal{E}_{emp}(\mathbf{w}, \phi(S))}{\partial w_{s,t} \partial w_{s',t'}} \right]_{1 \leq t,t' \leq m}.
$$

(8)

**Definition 3.** *For a model $f(x, \mathbf{w}) = g(\mathbf{w}\phi(x))$, $\mathbf{w} \in \mathbb{R}^{d \times m}$, with a twice differentiable function $g$, a twice differentiable loss function $\ell$ and a sample set $S$, relative flatness is defined by*

$$
\kappa_{Tr}^{\phi}(\mathbf{w}) := \sum_{s,s'=1}^{d} \langle \mathbf{w}_s, \mathbf{w}_{s'} \rangle \cdot Tr(H_{s,s'}(\mathbf{w}, \phi(S))),
$$

(9)

*where $Tr$ denote the trace and $\langle \mathbf{w}_s, \mathbf{w}_{s'} \rangle = \mathbf{w}_s \mathbf{w}_{s'}^T$ the scalar product of two row vectors.*

**Properties of relative flatness** (i) Relative flatness simplifies to ridge regression for linear models $f(x, \mathbf{w}) = \mathbf{w}x \in \mathbb{R}$ ($\mathcal{X} = \mathbb{R}^d$, $g = id$ and $\phi = id$) and squared loss: To see this, note that for any loss function $\ell$, the second derivatives with respect to the parameters $\mathbf{w} \in \mathbb{R}^d$ computes to $\frac{\partial^2 \ell}{\partial w_i \partial w_j} = \frac{\partial^2 \ell}{\partial (f(x,\mathbf{w}))^2} x_i x_j$. For $\ell(\hat{y}, y) = ||\hat{y} - y||^2$ the squared loss function, $\partial^2 \ell / \partial \hat{y}^2 = 2$ and the Hessian is independent of the parameters $\mathbf{w}$. In this case, $\kappa_{Tr}^{id} = c \cdot ||\mathbf{w}||^2$ with a constant $c = \sum_{x \in S} 2Tr(xx^T)$, which is the well-known Tikhonov (ridge) regression penalty.

(ii) Invariance under reparameterization: We consider neural network functions

$$
f(x) = \mathbf{w}^L \sigma(\dots \sigma(\mathbf{w}^2 \sigma(\mathbf{w}^1 x + b^1) + b^2) \dots) + b^L
$$

(10)

of a neural network of $L$ layers with nonlinear activation function $\sigma$. By letting $\phi^l(x)$ denote the composition of the first $l - 1$ layers, we obtain a decomposition $f(x, \mathbf{w}^l) = g^l(\mathbf{w}^l \phi^l(x))$ of the network. Using (9) we obtain a relative flatness measure $\kappa_{Tr}^l(\mathbf{w})$ for the chosen layer.

For a well-defined Hessian of the loss function, we require the network function to be twice differentiable. With the usual adjustments (equations only hold almost everywhere in parameter space), we can also consider neural networks with ReLU activation functions. In this case, Dinh et al. [5] noted that the network function —and with it the generalization performance— remains unchanged under linear reparameterization, i.e., multiplying layer $l$ with $\alpha > 0$ and dividing layer $k \neq l$ by $\alpha$, but common measures of the loss Hessian change. Our measure fixes this issue in relating flatness to generalization since the change of the loss Hessian is compensated by multiplication with the scalar products of weight matrices and is therefore invariant under layer-wise reparameterizations [cf. 26]. It is also invariant to neuron-wise reparameterizations, i.e., multiplying all incoming weights into a neuron by a positive number $\alpha$ and dividing all outgoing weights by $\alpha$ [23], except for neuron-wise reparameterizations of the feature layer $\phi^l$. Using a simple preprocessing step (a neuron-wise reparameterization with the variance over the sample), our proposed measure becomes independent of all neuron-wise reparameterizations.

**Theorem 4.** *Let $\sigma_i$ denote the variance of the $i$-th coordinate of $\phi^l(x)$ over samples $x \in S$ and $V = diag\left(\sigma_1, \ldots, \sigma_{n_{l-1}}\right)$. If the relative flatness measure $\kappa_{Tr}^l$ is applied to the representation*

$$f(x) = \mathbf{w}^L \sigma(\ldots \sigma(\mathbf{w}^l V \; \sigma(V^{-1}\mathbf{w}^{l-1}\sigma(\ldots \sigma(\mathbf{w}^1 x + b^1))\ldots) + V^{-1}b^{l-1}) + b^l)\ldots) + b^L$$

*then $\kappa_{Tr}^l$ is invariant under all neuron-wise (and layer-wise) reparameterizations*

We now connect flatness with feature robustness: Relative flatness approximates feature robustness for a model at a local minimum of the empirical risk, when labels are approximately constant in neighborhoods of the training samples $(\phi(x), y) \in \phi(S)$ in feature space.

**Theorem 5.** *Consider a model $f(x, \mathbf{w}) = g(\mathbf{w}\phi(x))$ as above, a loss function $\ell$ and a sample set $S$, and let $O_m \subset \mathbb{R}^{m \times m}$ denote the set of orthogonal matrices. Let $\delta$ be a positive (small) real number and $\mathbf{w} = \omega \in \mathbb{R}^{d \times m}$ denote parameters at a local minimum of the empirical risk on a sample set $S$. If the labels satisfy that $y(\phi_{\delta A}(x_i)) = y(\phi(x_i)) = y_i$ for all $(x_i, y_i) \in S$ and all $||A|| \leq 1$, then $f(x, \omega)$ is $((\delta, S, O_m), \epsilon)$-feature robust on average over $O_m$ for $\epsilon = \frac{\delta^2}{2m}\kappa_{Tr}^\phi(\omega) + \mathcal{O}(\delta^3)$.*

Applying the theorem to Eq. 5 implies that if the data is representative, i.e., $\mathcal{E}_{Rep}^\phi(f, S, \Lambda_{\mathcal{A}_\delta}) \approx 0$ for the distribution $\mathcal{A}_\delta$ of Prop. 2, then $\mathcal{E}_{gen}(f(\cdot, \omega), S) \lesssim \frac{\delta^2}{2m}\kappa_{Tr}^\phi(\omega) + \mathcal{O}(\delta^3)$. The assumption on locally constant labels in Thm. 5 can be relaxed to approximately locally constant labels without unraveling the theoretical connection between flatness and feature robustness. Appendix B investigates consequences from even dropping the assumption of approximately locally constant labels.

## 5  Flatness and Generalization

Combining the results from sections 2–4, we connect flatness to the generalization gap when the distribution can be represented by smooth probability densities on a feature space with approximately locally constant labels. By approximately locally constant labels we mean that, for small $\delta$, the loss in $\delta||\phi(x_i)||$-neighborhoods around the feature vector of a training sample $x_i$ is approximated (on average over all training samples) by the loss for constant label $y(x_i)$ on these neighborhoods. This and the following theorem connecting flatness and generalization are made precise in Appendix D.4.

**Theorem 6** (*informal*). *Consider a model $f(x, \mathbf{w}) = g(\mathbf{w}\phi(x))$ as above, a loss function $\ell$ and a sample set $S$, let $m$ denote the dimension of the feature space defined by $\phi$ and let $\delta$ be a positive (small) real number. Let $\omega$ denote a local minimizer of the empirical risk on a sample set $S$. If the distribution $\mathcal{D}$ has a smooth density $p_\mathcal{D}^\phi$ on the feature space $\mathbb{R}^m$ with approximately locally constant labels around the points $x \in S$, then it holds with probability $1 - \Delta$ over sample sets $S$ that*

$$\mathcal{E}_{gen}(f(\cdot, \omega), S) \lesssim |S|^{-\frac{2}{4+m}} \left( \frac{\kappa_{Tr}^\phi(\omega)}{2m} + C_1(p_\mathcal{D}^\phi, L) + \frac{C_2(p_\mathcal{D}^\phi, L)}{\sqrt{\Delta}} \right)$$

*up to higher orders in $|S|^{-1}$ for constants $C_1, C_2$ that depend only on the distribution in feature space $p_\mathcal{D}^\phi$ induced by $\phi$, the chosen $|S|$-tuple $\Lambda_\delta$ and the maximal loss $L$.*

To prove Theorem 6 we bound both $\epsilon$-representativeness and feature robustness in Eq. 5. For that, the main idea is that the family of distributions considered in Proposition 2 has three key properties: (i) it

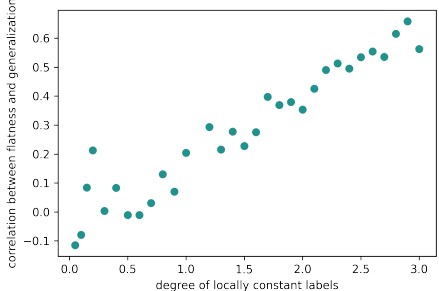
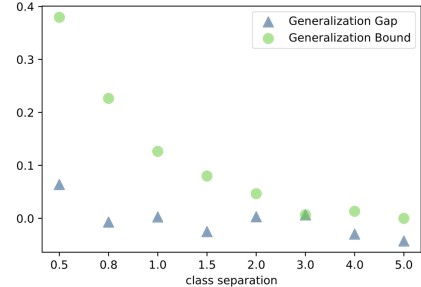

Figure 3: The correlation between flatness and generalization increases with the degree of locally constant labels.

Figure 4: Approximation of representativeness via KDE together with relative flatness leads to a tight generalization bound.

provides an explicit link between the distributions on feature matrices $\mathcal{A}_\delta$ used in feature robustness and the family of distributions $\Lambda_\delta$ of $\epsilon$-representativeness (Proposition 2) (ii) it allows us to bound feature robustness using Thm. 5; and (iii) it is simple enough that it allows us to use standard results of kernel density estimation (KDE) to bound representativeness.

Our bound suffers from the curse of dimensionality, but for the chosen feature space instead of the (usually much larger) input space. The dependence on the dimension is a result of using KDE uniformly over all distributions satisfying mild regularity assumptions. In practice, for a given distribution and sample set $S$, representativeness can be much smaller, which we showcase in a toy example in Sec. 6. In the so-called interpolation regime, where datasets with arbitrarily randomized labels can be fit by the model class, the obtained convergence rate is consistent with the no free lunch theorem and the convergence rate derived by Belkin et al. [2] for an interpolation technique using nearest neighbors.

A combination of our approach with prior assumptions on the hypotheses or the algorithm in accordance to statistical learning theory could potentially achieve faster convergence rate. Our herein presented theory is instead based solely on interpolation and aims to understand the role of flatness (a local property) in generalization: If the data is representative in feature layers and if the distribution can be approximated by locally constant labels in these layers, then flatness of the empirical risk surface approximates the generalization gap. Conversely, Equation 7 shows that flatness measures the performance under perturbed features only when labels are kept constant. As a result, we offer an explanation for the often observed correlation between flatness and generalization: Real-world data distributions for classification are benign in the sense that small perturbations in feature layers do not change the target class, i.e., they can be approximated by locally constant labels. (Note that the definition of adversarial examples hinges on this assumption of locally constant labels.) In that case, feature robustness is approximated by flatness of the loss surface. If the given data and its feature representation are further $\epsilon$-representative for small $\epsilon \approx 0$, then flatness becomes the main contributor to the generalization gap leading to their noisy, but steady, correlation.

## 6 Empirical Validation

We empirically validate the assumptions and consequences of the theoretical results derived above [2]. For that, we first show on a synthetic example that the empirical correlation between flatness and generalization decreases if labels are not locally constant, up to a point when they are not correlated anymore. We then show that the novel relative flatness measure correlates strongly with generalization, also in the presence of reparameterizations. Finally, we show in a synthetic experiment that while representativeness cannot be computed without knowing the true data distribution, it can in practice be approximated. This approximation—although technically not a bound anymore—tightly bounds the generalization gap. Synthetic data distributions for binary classification are generated by sampling 4 Gaussian distributions in feature space (two for each class) with a given distance between their means (class separation). We then sample a dataset in feature space $S^\phi$, train a linear classifier

---

[2] Code is available at https://github.com/kampmichael/relativeFlatnessGeneralization.

$\psi$ on the sample, randomly draw the weights of a 4-layer MLP $\phi$, and generate the input data as $S = (\phi^{-1}(S_x^\phi), S_y^\phi)$. This yields a dataset $S$ and a model $f = \phi \circ \psi$ such that $\phi(S)$ has a given class separation. Details on the experiments are provided in Appdx. C.

**Locally constant labels:** To validate the necessity of locally constant labels, we measure the correlation between the proposed relative flatness measure and the generalization gap for varying degrees of locally constant labels, as measured by the class separation on the synthetic datasets. For each chosen class separation, we sample 100 random datasets of size 500 on which we measure relative flatness and the generalization gap. Fig. 3 shows the average correlation for different degrees of locally constant labels, showing that the higher the degree, the more correlated flatness is with generalization. If labels are not locally constant, flatness does not correlate with generalization.

**Approximating representativeness:** While representativeness cannot be calculated without knowing the data distribution, it can be approximated from the training sample $S$ by the error of a density estimation on that sample. For that, we use multiple random splits of $S$ into a training set $S_{\text{train}}$ and a test set $S_{\text{test}}$, train a kernel density estimation on $S_{\text{train}}$ and measure its error on $S_{\text{test}}$. Again, details can be found in Appx. C. The lower the class separation of the synthetic datasets, the harder the learning problem and the less representative a random sample will be. For each sample and its distribution, we compute the generalization gap and the approximation to the generalization bound. The

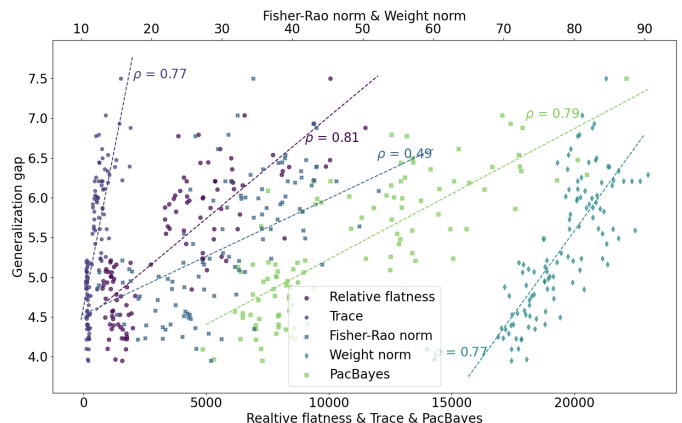

Figure 5: The generalization gap for various local minima correlates stronger with relative flatness than standard flatness, Fisher-Rao norm, PacBayes based measure and weights norm (points corresp. to local minima).

results in Fig. 4 show that the approximated generalization bound tightly bounds the generalization error (note that this approximation is technically not a bound anymore). Moreover, as expected, the bound decreases the easier the learning problems become.

**Relative flatness correlates with generalization:** We validate the correlation of relative flatness to the generalization gap in practice by measuring it for 110 different local minima—achieved via different learning setups, such as initialization, learning rate, batch size, and optimization algorithm— of LeNet5 [19] on CIFAR10 [18]. We compare this correlation to the classical Hessian-based flatness measures using the trace of the loss-Hessian, the Fisher-Rao norm [21], the PACBayes flatness measure that performed best in the extensive study of Jiang et al. [14] and the $L_2$-norm of the weights. The results in Fig. 5 show that indeed relative flatness has higher correlation than all the competing measures. Of these measures, only the Fisher-Rao norm is reparameterization invariant but shows the weakest correlation in the experiment. In Appdx C we show how reparameterizations of the network significantly reduce the correlation for non-reparameterization invariant measures.

## 7 Discussion and Conclusion

Contributing to the trustworthiness of machine learning, this paper provides a rigorous connection between flatness and generalization. As to be expected for a local property, our association between flatness and generalization requires the samples and its representation in feature layers to be representative for the target distribution. But our derivation uncovers a second, usually overlooked condition. Flatness of the loss surface measures the performance of a model close to training points when labels are kept locally constant. If a data distribution violates this, then flatness cannot be a good indicator for generalization.

Whenever we consider feature representations other than the input features, the derivation of our results makes one strong assumption: the existence of a target output function $y(\phi(x))$ on the feature space $\phi(\mathcal{X})$. By moving assumptions on the distribution from the input space to the feature space, we achieve a bound based on interpolation that depends on the dimension of the feature layer instead of the input space. Hence, we assume that the feature representation is reasonable and does not lose information that is necessary for predicting the output. To achieve faster convergence rates independent of any involved dimensions, future work could aim to combine our approach of interpolation with a prior-based approach of statistical learning theory.

Our measure of relative flatness may still be improved in future work. Better estimates for the generalization gap are possible by improving the representativeness of local distributions in two ways: The support shape of the local distributions can be improved and their volume-parameter $\delta$ can be optimally chosen. Both improvements will affect the derivation of the measure of relative flatness as an estimation of feature robustness for the corresponding distributions on feature matrices. Whereas different support shapes change the trace to a weighted average of the Hessian eigenvalues, the volume parameter can provide a correcting scaling factor. Both approaches seem promising to us, as our relative measure from Definition 3 already outperforms the competing measures of flatness in our empirical validation.

## Acknowledgements

Cristian Sminchisescu was supported by the European Research Council Consolidator grant SEED, CNCS-UEFISCDI (PN-III-P4-ID-PCE-2016-0535, PN-III-P4-ID-PCCF-2016-0180), the EU Horizon 2020 grant DE-ENIGMA (688835), and SSF.

Mario Boley was supported by the Australian Research Council (under DP210100045).

We would like to thank Julia Rosenzweig, Dorina Weichert, Jilles Vreeken, Thomas Gärtner, Asja Fischer, Tatjana Turova and Alexandru Aleman for the great discussions.

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
