**Organization of the Appendix**  The appendix is organized as follows:

**A – Related work** contains an extended discussion on related work.

**B – The effect of local label changes** discusses consequences for the association of flatness and generalization for general output function $y(x)$ without the assumption of locally constant labels.

**C – Details on the Empirical Validation** contains a detailed description of the experiments.

**D – Proofs** contains the full proofs to all statements. In detail:

**E – Relative flatness for a uniform bound over general distributions on feature matrices** defines a variant of relative flatness that uniformly bounds feature robustness over all feature matrices.

## A   Related Work

It has long been observed that algorithms searching for flat minima of the loss curve lead to better generalization [10, 11]. More recently, an association between flatness and low generalization error has also been validated empirically in deep learning [16, 27, 38]. Here, flatness is measured by the Hessian of the empirical loss evaluated at the model at hand. Indeed, in their recent extensive empirical study of generalization measures, Jiang et al. [14] found that measures based on flatness have the highest correlation with generalization.

For models trained with stochastic gradient descent (SGD), this could present a (partial) explanation for their generalization performance, since the convergence of SGD can be connected to flat local minima by studying SGD as an approximation of a stochastical differential equation [13, 43]. However, while large and small batch methods appear to converge in different basins of attraction, the basins can be connected by a path of low loss, i.e., they can actually converge into the same basin [32]. Moreover, as Dinh et al. [5] remarked, classical flatness measures—which are based only on the Hessian of the loss function—cannot theoretically be related to generalization: For deep neural networks with ReLU activation functions, there are linear reparameterizations that leave the network function unchanged (hence, also the generalization performance), but change any measure derived only from the loss Hessian. Novel measures related to flatness have been proposed that are invariant to linear reparameterizations [21, 31, 37]. Rangamani et al. [31] measure flatness in the quotient space of a suitable equivalence relation, and Liang et al. [21] utilize the Fisher-Rao metric, but the theoretical connection of these two measures to generalization is not well-understood. Neyshabur et al. [26] noted that the reparamterization-issue can in general be resolved by balancing a measure of flatness with a norm on the parameters, which is the way that normalized flatness [37], Fisher-Rao metric [21] and our proposed relative flatness become reparameterization-invariant. However, the solution proposed in Neyshabur et al. [26] necessitates data-dependent priors [7] or related approaches, which "adds non-trivial costs to the generalization bounds" [37].

The question arises in which way the loss Hessian and parameter norm should be combined. A simple scaling of the full Hessian with the squared parameter norm does not provide a reparamterization-invariant measure. Doing so for each layer independently and summing up the results provides a measure that is only invariant under layer-wise reparameterizations. Similarly, only considering a single feature layer yields a measure that is layer-wise reparameterization invariant [30]. While the resulting measure can also be analyzed within our framework to obtain a bound on feature robustness, our proposed measure yields a tighter bound and is also invariant under neuron-wise reparameterizations.

Tsuzuku et al. [37] derive a flatness measure that scales an approximation to the loss Hessian by a parameter-dependent term. Their proposed measure correlates well with generalization and is theoretically connected to it via the PAC-Bayesian framework. However, this connection requires the assumption of Gaussian priors and posteriors and is not informative with respect to conditions under which this connection holds. Moreover the measure is impractical, since computing it requires

solving an optimization problem for every layer that can be numerically unstable. (Tsuzuku et al. [37] propose a solution to the numerical instability at the cost of losing the reparameterization-invariance.) Instead, relative flatness can be computed directly and takes only parameters of a specific layer into account—although combining relative flatness of all layers by simple summation is possible.

A series of recent papers studies flatness by minimizing the loss at local perturbations of the parameters considering $\min_{\mathbf{a}} \mathcal{E}_{emp}(f(\mathbf{w} + \mathbf{a}), S)$ [8, 34, 39, 44]. Regularization techniques enforcing these notions of flatness during training in classification tasks lead to better generalization. These empirical results follow earlier works by Chaudhari et al. [4] and Izmailov et al. [12] that similarly obtained better generalization by enforcing flatter minima. Their observations are well-explained by our theory: Low error at perturbations $\mathcal{E}_{emp}(f(\mathbf{w} + \mathbf{a}), S)$ lead to good generalization around training samples. This requires that the underlying distribution has (approximately) locally constant labels (using key equation (1)), which is reasonable for the image classification tasks they consider.

Xu and Mannor [40] propose a notion of robustness over a partion of the input space and derive generalization bounds based on it. However, their notion requires the choice of a partitioning of the input space before seeing any samples. Thus, robustness over the partition can be hard to estimate for a model that depends on a sample set $S$. Our notion of feature robustness is measured around a given sample set and thus does not require a uniform data-independent partitioning. Such a sample-dependent notion of robustness is necessary to connect it to the flatness of the loss surface, since flatness is a local property around training points.

Novak et al. [27] find that robustness to input perturbation as measured by the input-output Jacobian correlates well with generalization on classification tasks. This is in line with our findings applied to $\phi = id_{\mathcal{X}}$ chosen as the identity (for neural networks this means considering the input layer as features): it follows from Equation 1 that robustness to input perturbations directly relates to flatness. Therefore, these findings give additional empirical evidence to the correlation between flatness and generalization. Yao et al. [41] study the Hessian with respect to the input $x \in \mathcal{X}$ and also find that robust learning tends to converge to minima where the input-ouptut Hessian has small eigenvalues.

# B The effect of local label changes

For classification tasks with one-hot vectors as labels, the assumption of locally constant labels, i.e., locally constant target output function $y(x)$, seems reasonable since we would not expect the class label to change under (infinitesimally) small changes. One could nonetheless consider a smooth output function with values encoding class probabilities for classification, which may change locally around the training points. For regression tasks, the assumption of locally constant output function is rather unrealistic or at the very least restrictive.

Taking the term defining feature robustness (4) as a starting point, we investigate its connection to flatness when the output function $y(x)$ is a smooth function. In the usual setting of machine learning, this information is unknown. We will show that label changes can contribute stronger to the loss in neighborhoods around training samples than (relative) flatness.

To investigate the label dependence, we use the same trick as in (7) to transfer perturbations in the input $x$ to perturbations in parameter space $\mathbf{w}$. To simplify the analysis, we apply feature robustness to the input space (i.e., we only consider $\phi = id_{\mathcal{X}}$ here). Let $f(x, \mathbf{w}) = \psi(\mathbf{w}x)$ be a model composed of a matrix multiplication of $x$ with $\mathbf{w}$ and a differentiable predictor function $\psi$.

$$
\begin{aligned}
\mathcal{E}_{\tilde{F}}(f, S, A) &= \frac{1}{n} \sum_{i=1}^{n} \left( \ell(f(x_i + \delta A x_i, \mathbf{w}), y[x_i + \delta A x_i]) - \ell(f(x_i, \mathbf{w}), y_i) \right) \\
&= \frac{1}{n} \sum_{i=1}^{n} \left( \ell(f(x_i, \mathbf{w} + \delta \mathbf{w} A), y[x_i + \delta A x_i]) - \ell(f(x_i, \mathbf{w}), y_i) \right)
\end{aligned}
$$

Defining a function

$$
\gamma_i(\delta) = \ell(f(x_i, \mathbf{w} + \delta \mathbf{w} A), y[x_i + \delta A x_i]), \tag{8}
$$

we have that $\mathcal{E}_{\tilde{F}}(f, S, A) = \frac{1}{n} \sum_{i=1}^{n} \gamma_i(\delta)$. For each $\gamma_i$ we use Taylor approximation in $\delta$. In the following, we write $\ell_{\mathbf{w}}(x_i, \mathbf{w}^*, y_i)$ for the first derivative of the loss with changes in $\mathbf{w}$ at $x_i, y_i = y[x_i]$ and $\mathbf{w}^*$, and we write $\ell_y(x_i, \mathbf{w}^*, y_i)$ for the first derivative of the loss with changes of the output $y$ at $x_i, y_i = y[x_i]$ and $\mathbf{w}^*$. Similarly, we consider second derivatives $\ell_{\mathbf{ww}}, \ell_{yy}$ and $\ell_{\mathbf{w}y}$. Finally, we denote the derivative of $y(x)$ with respect to $x$ by $y_x$ and the second derivative by $y_{xx}$. Then,

$$
\gamma_i'(0) = \ell_{\mathbf{w}}(x_i, \mathbf{w}^*, y_i) \cdot (\mathbf{w}^* A) + \ell_y(x_i, \mathbf{w}^*, y_i) \cdot (y_x(x_i) \cdot A x_i) \tag{9}
$$

and

$$
\gamma_i''(0) = (\mathbf{w}^* A)^T \ell_{\mathbf{ww}}(x_i, \mathbf{w}^*, y_i)(\mathbf{w}^* A) + (y_x(x_i) \cdot A x_i)^T \ell_{yy}(x_i, \mathbf{w}^*, y_i)(y_x(x_i) \cdot A x_i) \tag{10}
$$

$$
+ \sum_{\text{labels } c} \ell_{y_c}(x_i, \mathbf{w}^*, y_i) \cdot (A x_i)^T (y_c)_{xx}(x_i) \cdot (A x_i) + 2(y_x(x_i) \cdot A x_i)^T \ell_{y\mathbf{w}}(x_i, \mathbf{w}^*, y_i)(\mathbf{w}^* A) \tag{11}
$$

At a critical point we have that $\sum_i \ell_{\mathbf{w}}(x_i, \mathbf{w}^*, y_i) = 0$, but since we do not know how the target output function $y(x)$ changes locally, we do not necessarily[3] enforce that $\sum_i \ell_y(x_i, w^*, y_i) = 0$ at a local optimum. In that case, $\mathcal{E}_{\tilde{F}}(f, S, A) = \sum_i \ell_y(x_i, w^*, y_i)\delta + \mathcal{O}(\delta^2)$ has a non-zero term of first order in $\delta$ and flatness only contributes as a term of order two. Similarly, other terms in (10) can be nonzero, further reducing the influence of relative flatness to a bound on feature robustness.

As an interesting special case, we note that for one-hot encoded labels in classification and letting the output function $y(x)$ describe a parameter vector of a conditional label-distribution given $x$, we have $y_x(x_i) = 0$ (recall that we suppose $y(x_i) = y_i$) as each vector component is either 1 or 0 and must be a local extreme point ($y(x)$ cannot contain values larger than 1 or smaller than 0 by assumption),

We leave a detailed investigation of the consequences of label changes as future work, but identify the implicit assumption of locally constant labels in loss Hessian-based flatness measures as a possible limitation: Flatness can only be descriptive if optimal label changes are approximately locally constant. The fact that a strong correlation between flatness and generalization gap has been often observed points to the fact that distributions in practice satisfy this implicit assumption.

---

[3]This depends on the loss function in use.

## C Details on the Empirical Validation

Here we provide additional details on the empirical evaluation. Jupyter notebooks containing the experiments are available at https://github.com/kampmichael/relativeFlatnessGeneralization, ensuring reproducibility, together with an implementation of the relative flatness measure in pytorch [28].

### C.1 Synthetic Experiments

The experiments on *locally constant labels* and *approximating representativeness* use a synthetic sample in feature space. The schema for both experiments is to

1. create a synthetic dataset in feature space $S^\phi$ and test set $T^\phi$,
2. create a model $f = \phi \circ \psi$,
3. derive input data as $S = \left(\phi^{-1}\left(S_x^\phi\right), S_y^\phi\right), T = \left(\phi^{-1}\left(T_x^\phi\right), T_y^\phi\right)$
4. compute relative flatness (or other measures) of $f$ on $S$,
5. and estimate its generalization gap by computing the empirical risk of $f$ on $S$, and computing the test error on the test test $T$ to estimate the risk.

1) To create $S^\phi$ with a given class separation $c$, we randomly sample 4 cluster centroids $\theta$ from a hypercube in $\mathbb{R}^6$ and scale them so that their distance is $c$. We then sample a random covariance matrix $\Sigma$ for each cluster and sample points from a Gaussian $\mathcal{N}(\theta, \Sigma)$. Furthermore, we create two redundant features that are a random linear combination of the 6 informative features. We obtain labels by assigning two clusters to class 1 and the other two to class $-1$.

2) We create the model $f$ by first training a linear model $\psi$ on $S^\phi$ using ridge regression from scikit-learn [29]. We then sample a random 4-layer MLP (with architecture 784-512-128-16-8, tanh activation, and Glorot initialization [9]) that we use as feature extractor $\phi$. With this, we obtain the 5-layer MLP $f = \phi \circ \psi$ by adding an $8 - 2$ layer with weights obtained from $\psi$.

3) We obtain input data $S$ by reverse propagation of samples in feature space $S_x^\phi$ through the 4-layer MLP $\phi$. This is an approximation to the inverse feature extractor $\phi^{-1}$. For the output of each layer $z$, we first compute $z' = tanh^{-1}(z)$, i.e., the inverse of the activation function. We then solve $Wz + b = x$, where $W, b$ are the weights and bias of that layer, and $x$ is the corresponding input we want to compute. This yields $S_x = \phi^{-1}(S_x^\phi)$. Note that this reverse propagation of samples introduces a small error. To keep experiments realistic, we discard $S^\phi$ after this step and use only the input dataset $S$ and model $f$ in our computations.

4) We compute relative flatness as in Def. 3 (an implementation in pytorch is available on github, see above).

5) We compute the empirical risk of $f$ on $S$ and estimate the risk on a test set. For the experiments on locally constant labels, generate 5000 samples, use a training set of size 500, a test set of size 4500 (to ensure an accurate estimate of the risk), and repeat the experiment 100 times for each class separation $c$. For the experiment on approximating representativeness, we use a sample of size 600 and perform 3-fold cross-validation.

**Locally constant labels:** For classification, labels are locally constant if in a neighborhood around each point the label does not change. They are approximately locally constant, if this holds for most points. By increasing the distance between the means of the Gaussians, we decrease the likelihood of a point within a neighborhood having a different label. For a finite sample, this means that the likelihood of observing two points close by with different labels decreases. Thus, by increasing the class separation parameter, we increase the degree of locally constant labels.

**Approximating representativeness:** A finite random sample as described in 1) has a higher chance of being representative when the means of the Gaussians have a high distance, because each individual Gaussian can be interpolated easily. Of course, the actual representativeness of a sample at hand can vary. Note that this is a very simple form of generating datasets with varying "difficulty". It will be interesting to further explore the impact of the choice of data distribution on (an approximation to) representativeness.

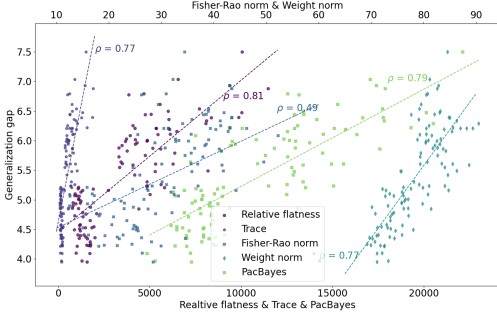
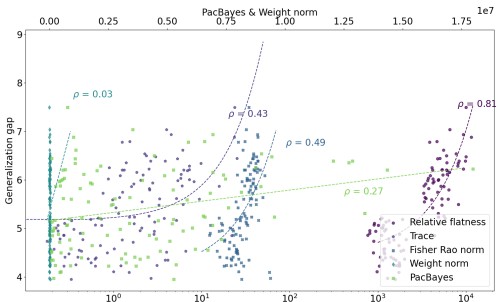

Figure 6: Generalization gap and various flatness measures for 110 local minima as presented in Fig. 5. The generalization gap correlates stronger with relative flatness than standard flatness, Fisher-Rao norm, a PAC-Bayes based measure and the weights norm.

Figure 7: Modifying the local minima in the plot Fig. 6 by reparameterization shows that the proposed relative flatness and the Fisher-Rao norm are invariant to them. It furthermore shows a strong decline in correlation for all other measures.

Experiments on the synthetic datasets are run on a laptop with Intel Core i7 and NVIDIA GeForce GTX 965 M 2 GB GPU. The code of the experiments is provided as a jupyter notebook so that they can be easily reproduced.

## C.2 Relative Flatness Correlates with Generalization

In this experiment, we validate that the proposed relative flatness correlates strongly with generalization in practice. For that, we measure relative flatness (as well as classical flatness measured by the trace of the loss Hessian, the Fisher-Rao norm, a PAC-Bayes based measure [4], and the weight norm) together with the generalization gap for various local minima.

To obtain model parameters at various local minima, we train networks (LeNet5 [20]) on the CIFAR10 dataset until convergence (measured in terms of achieving a loss of less than 0.1 during an epoch, which has been used as a criteria for convergence in similar experiments [14]) with varying hyperparameters. In accordance to works studying the impact of hyperparameters on generalization [14, 16, 27, 31, 38], we vary learning rate, mini-batch size, initialization, and optimizer. We vary the mini batch size in $64, 128, 256, 512, 1024$, and the learning rate in $0.0001, 0.02, 0.05$, running 10 randomly initialized training

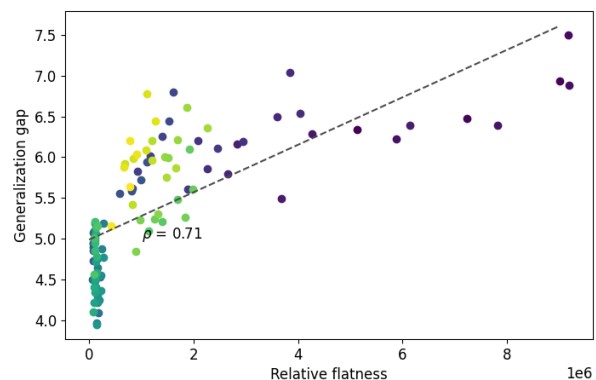

Figure 8: The generalization gap for various local minima correlates with relative flatness measured on the layer different from penultimate layer.

rounds for each setup. We use SGD, ADAM, and RMSProp as optimizers. We only use combinations that lead to convergence. The experiments were conducted on a cluster node with 4 NVIDIA GPU GM200 (GeForce GTX TITAN X). As discussed in Sec. 6, relative flatness has the highest correlation with generalization from all measures we analyzed.

[4]The implementation of PAC-Bayes based flatness measure is taken from https://github.com/nitarshan/robust-generalization-measures/blob/master/data/generation/measures.py

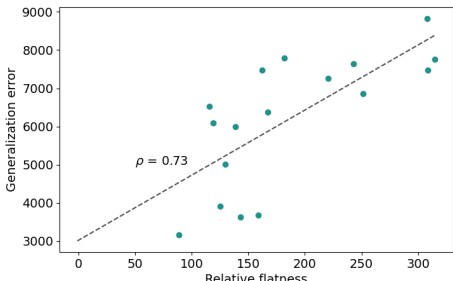

Figure 9: Layer1-based relative flatness for MNIST experiment. Layer1 is the topmost layer.

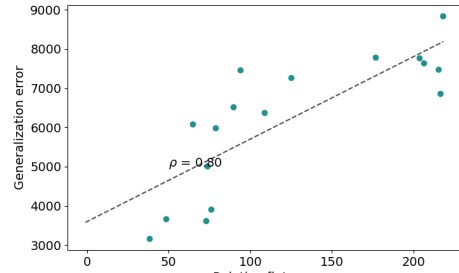

Figure 10: Layer2-based relative flatness for MNIST experiment.

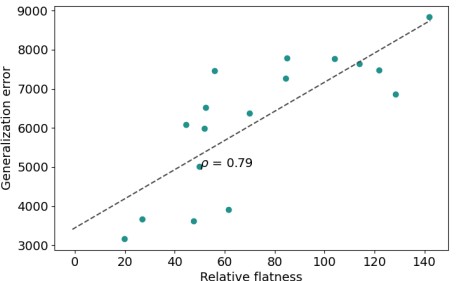

Figure 11: Layer3-based relative flatness for MNIST experiment.

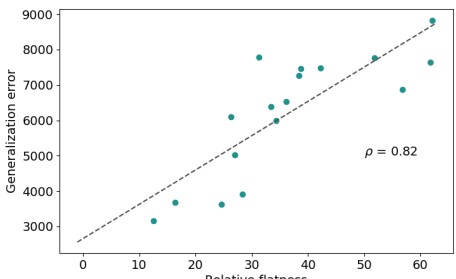

Figure 12: Layer4-based relative flatness for MNIST experiment. Layer4 is penultimate.

To study the effect of reparameterization, we apply layer-wise reparameterizations on the trained network using random factors in the interval $[5, 25]$ which yields a set of novel local minima. The results in Fig. 7 show that both our proposed relative flatness and the Fisher-Rao norm are invariant to these reparameterization. For all other measures, the correlation with generalization declines substantially. The same would hold for neuron-wise reparameterizations, since both relative flatness and the Fisher-Rao norm are also neuron-wise reparameterization invariant. Relative flatness and the Fisher-Rao norm are also invariant under neuron-wise reparameterizations, which could be used to further break the correlation for the other measures. For future work it would be interesting to investigate further symmetries in neural networks and the impact of reparameterizations along these symmetries on flatness measures.

In addition to the calculation of the relative flatness using the feature space of the penultimate layer, we also performed calculations for another fully-connected layer in the network. The resulting correlation can be seen in Fig. 8. It keeps the high correlation value, but due to less optimal feature space we observe smaller number, than in the previous calculation. Nevertheless, it demonstrates that any $\phi$-$\psi$ separation allows to compute relative flatness.

For checking deeper the viability of the relative flatness computed in different feature representations, we ran a similar experiment with a fully-connected $(784 - 50 - 50 - 50 - 30 - 10)$ neural network trained on MNIST dataset (Fig. 9, 10, 11, 12). We varied parameters of the optimization (batch size in $1000 - 2000 - 4000 - 8000$ and learning rate in $0.02 - 0.04 - 0.08 - 0.16$ in order to keep the ratio between them constant) and trained each network with SGD for $500$ epochs. Only the networks that achieved training loss lower than $0.07$ are used for the plots. The observed correlation with generalization gap is high for each of the representations in the network.

# D   Proofs

## D.1   Proof of Proposition 2

*Proof.* Let $\mathcal{K}_h$ denote probability distribution defined by a rotational-invariant kernel $k_h$ as in (6) with $k_h(0, z) = \frac{1}{h^m} \cdot k\left(\frac{||z||}{h}\right) \cdot \mathbb{1}_{||z|| < h}$ and let $\lambda_i(z) = k_{\delta||\phi(x_i)||}(0, z)$. Let $\mathcal{L}$ denote a continuous function on $\mathbb{R}^m$ and $\mathcal{O}_m$ the set of orthogonal matrices in $\mathbb{R}^{m \times m}$. We show that there exists a probability measure $\kappa$ on a set $M_\delta$ of matrices of norm smaller than $\delta$, defining a probability distribution $\mathcal{A}_\delta$, and a probability measure $\omega$ on the product space $(0, \delta] \times \mathcal{O}_m$ such that for each $z \in \mathbb{R}^m \setminus \{0\}$:

$$\mathbb{E}_{A \sim \mathcal{A}_\delta}\left[\mathcal{L}(z + Az)\right] = \mathbb{E}_{(r,O) \sim \omega}\left[\mathcal{L}(z + rOz)\right] = \mathbb{E}_{\zeta \sim \mathcal{K}_{\delta||z||}}\left[\mathcal{L}(z + \zeta)\right] \tag{12}$$

Applying this result for each $i = 1, \ldots, |S|$ to $\mathcal{L}_i(z) = \ell(\psi(\mathbf{w}, z), y_i[z])$ at $z = \phi(x_i)$ completes the proof. For all the standard measure-theoretic concepts used in the proof, we refer the reader to [17].

Fix some $\zeta_0$ in $\mathbb{R}^m$ with $||\zeta_0|| = 1$. We consider the Haar measure $\mu$ on the set of orthogonal matrices $\mathcal{O}_m$. By [17, Proposition 3.2.1] and the change of variables formula, we have for each $r \in (0, \delta]$

$$\int_{O \in \mathcal{O}_m} \mathcal{L}(z + r||z||O\zeta_0)\, d\mu(O) = \frac{1}{\text{Vol}(S^{m-1})} \int_{\xi \in S^{m-1}} \mathcal{L}(z + r||z||\xi)\, d\xi$$

where $S^{m-1}$ is the $(m-1)$-sphere. We multiply both sides by $\frac{\text{Vol}(S^{m-1})}{\delta^m} k\left(\frac{r}{\delta}\right) r^{m-1}$, integrate over $r \in (0, \delta]$ to obtain

$$\frac{\text{Vol}(S^{m-1})}{\delta^m} \int_{r=0}^{\delta} \int_{O \in \mathcal{O}_m} \mathcal{L}(z + r||z||O\zeta_0) k\left(\frac{r}{\delta}\right) r^{m-1} dr d\mu(O)$$

$$= \frac{1}{\delta^m} \int_{r=0}^{\delta} \int_{\xi \in S^{m-1}} \mathcal{L}(z + r||z||\xi) k\left(\frac{r}{\delta}\right) r^{m-1}\, dr\, d\xi$$

$$= \frac{1}{\delta^m} \int_{||\zeta|| \leq \delta} \mathcal{L}(z + ||z||\zeta) k\left(\frac{||\zeta||}{\delta}\right)\, d\zeta$$

$$= \int_{||\zeta|| \leq \delta||z||} \mathcal{L}(z + \zeta) \frac{1}{(\delta||z||)^m} k\left(\frac{||\zeta||}{\delta||z||}\right)\, d\zeta$$

Introducing the product measure $\omega := \frac{\text{Vol}(S^{m-1})}{\delta^m} \cdot \left(k\left(\frac{r}{\delta}\right) r^{m-1} dr \times \mu\right)$ on $(0, \delta] \times \mathcal{O}_n$, this implies that

$$\mathbb{E}_{(r,O) \sim \omega}\left[\mathcal{L}(z + r||z||O\xi_0)\right] = \mathbb{E}_{\zeta \sim \mathcal{K}_{\delta||z||}}\left[\mathcal{L}(z + \zeta)\right] \tag{13}$$

The measure $\omega$ can be pushed forward to a measure on matrices of norm $||A|| \leq \delta$. For this, consider the homeomorphism

$$H : (0, \delta] \times \mathcal{O}_n \to \{rO \mid r \in (0, \delta], O \in \mathcal{O}_n\} =: M_\delta \subseteq \{A \in \mathbb{R}^{n \times n} \mid ||A|| \leq \delta\}$$

given by $H(r, O) = rO$. We use the inverse of $H$ to push forward the measure $\omega$ to a measure $\kappa$ on $M_\delta$ and obtain from (13) that

$$\mathbb{E}_{A \sim (M_\delta, \kappa)}\left[\mathcal{L}(z + ||z||A\zeta_0)\right] = \mathbb{E}_{\zeta \sim \mathcal{K}_{\delta||z||}}\left[\mathcal{L}(z + \zeta)\right]$$

Finally, there exists an orthogonal matrix $O$ such that $O||z||\zeta_0 = z$. Since $\kappa(A) = \kappa(AO^{-1})$ by definition of $\kappa$ and since $M_\delta O = M_\delta$, we get for any $z$ that

$$\mathbb{E}_{\zeta \sim \mathcal{K}_{\delta||z||}}\left[\mathcal{L}(z + \zeta)\right] = \mathbb{E}_{A \sim (M_\delta, \kappa)}\left[\mathcal{L}(z + A||z||\zeta_0)\right]$$

$$= \mathbb{E}_{A \sim (M_\delta O^{-1}, \kappa)}\left[\mathcal{L}(z + AO||z||\zeta_0)\right]$$

$$= \mathbb{E}_{A \sim (M_\delta, \kappa)}\left[\mathcal{L}(z + Az)\right]$$

Hence, the probability distribution $\mathcal{A}_\delta$ on matrices with norm bounded by $\delta$ defined by the probability measure $\kappa$ with support on $M_\delta$, and the space $(0, \delta] \times \mathcal{O}_m$ equipped with $\omega = \frac{\text{Vol}(S^{m-1})}{\delta^m} \cdot \left(k\left(\frac{r}{\delta}\right) r^{m-1} dr \times \mu\right)$ give the desired probability distributions satisfying (12). $\qquad \square$

### D.2 Proof of Theorem 4

We rephrase Theorem 4 split into Theorem 7 and a subsequent corollary that specify the reparameterizations under consideration. Let $f = f(\mathbf{w}^1, b^1, \mathbf{w}^2, b^2, \ldots, \mathbf{w}^L, b^L)$ denote a ReLU network function parameterized by parameters $\mathbf{w}^k = w_{s,t}^k$ and bias $b^k = b_s^k$ of the $k$-th layer given by

$$f(x) = \mathbf{w}^L \sigma(\ldots \sigma(\mathbf{w}^l \ \sigma(\mathbf{w}^{l-1}\sigma(\ldots \sigma(\mathbf{w}^1 x + b^1))\ldots) + b^{l-1}) + b^l)\ldots) + b^L.$$

Recall that we let $\phi^l(x)$ denote the composition of the first $(l-1)$ layers so that we obtain a decomposition $f(x, \mathbf{w}^l) = g^l(\mathbf{w}^l \phi^l(x))$ of the network. Using (9) we obtain a relative flatness measure $\kappa_{Tr}^l(\mathbf{w})$ for the chosen layer.

A layer-wise reparameterizaton multiplies all weights in a layer $l$ with a positive number $\lambda$ and divides the weights of another layer $l' \neq l$ by the same $\lambda$. Due to the positive homogeneity of the ReLU activation, this does not change the network function. By a neuron-wise reparameterization, we mean the operation that multiplies all weights into a neuron by some positive $\lambda$ and divides all outgoing weights of the same neuron by $\lambda$. Again, the positive homogeneity of the activation function implies that this operation does not change the network function. A layer-wise reparameterization is simply the parallel application of neuron-wise reparameterization for all neurons of one layer with the same reparameterization parameter $\lambda > 0$.

**Theorem 7.** *Let $f = f(\mathbf{w}^1, b^1, \mathbf{w}^2, b^2, \ldots, \mathbf{w}^L, b^L)$ denote a neural network function parameterized by parameters $\mathbf{w}^k = w_{s,t}^k$ and bias $b^k = b_s^k$ of the $k$-th layer. Suppose there are positive numbers $\lambda_{s,t}^k$ such that the parameters $\mathbf{w}_\lambda^k, b_\lambda^k$, obtained from multiplying $w_{s,t}^k$ at matrix position $(s,t)$ in layer $k$ by $\lambda_{s,t}^k$ and $b_s^k$ by $\lambda_{(s,0)}^k$, satisfy that $f(\mathbf{w}^1, b^1, \mathbf{w}^2, b^2, \ldots, \mathbf{w}^L, b^L) = f(\mathbf{w}_\lambda^1, b_\lambda^1, \mathbf{w}_\lambda^2, b_\lambda^2, \ldots, \mathbf{w}_\lambda^L, b_\lambda^L)$. If for the layer with index $l$ it holds that $\lambda_{(s,t)}^l = \lambda_{(s,t')}^l$ for each $s, t$ and $t'$, then $\kappa_{Tr}^l(\mathbf{w}) = \kappa_{Tr}^l(\mathbf{w}_\lambda)$ for the notion of relative flatness from Definition 3.*

**Corollary 8.** *Let $\sigma_i$ denote the variance of the $i$-th coordinate of $\phi^l(x)$ over samples $x \in S$ and $V = diag(\sigma_1, \ldots, \sigma_{n_{l-1}})$. If the relative flatness measure $\kappa_{Tr}^l$ is applied to the representation $f = f(\mathbf{w}^1, b^1, \ldots, V^{-1}\mathbf{w}^{l-1}, V^{-1}b^{l-1}, \mathbf{w}^l V, b^l, \mathbf{w}^{l+1}, b^{l+1} \ldots, \mathbf{w}^L, b^L)$, i.e.,*

$$f(x) = \mathbf{w}^L \sigma(\ldots \sigma(\mathbf{w}^l V \ \sigma(V^{-1}\mathbf{w}^{l-1}\sigma(\ldots \sigma(\mathbf{w}^1 x + b^1))\ldots) + V^{-1}b^{l-1}) + b^l)\ldots) + b^L,$$

*then $\kappa_{Tr}^l$ is invariant under all neuron-wise (and layer-wise) reparameterizations*

*Proof.* We are given a neural network function $f(x; \mathbf{w}^1, b^1, \ldots, \mathbf{w}^L, b^L)$ parameterized by parameters $\mathbf{w}^k$ and bias terms $b^k$ of the $k$-th layer and positive numbers $\lambda_{(s,t)}^1, \ldots, \lambda_{(s,t)}^L$ such that the parameters $\mathbf{w}_\lambda^k$ obtained from multiplying weight $w_{(s,t)}^k$ at matrix position $(s,t)$ in layer $k$ by $\lambda_{(s,t)}^k$ and $b_s^k$ by $\lambda_{(s,0)}^k$ satisfies that

$$f(x; \mathbf{w}^1, b^1, \mathbf{w}^2, b^2, \ldots, \mathbf{w}^L, b^L) = f(x; \mathbf{w}_\lambda^1, b_\lambda^1, \mathbf{w}_\lambda^2, b_\lambda^2 \ldots, \mathbf{w}_\lambda^L, b_\lambda^L)$$

for all $\mathbf{w}^k, b^k$ and all $x$.

For fixed layer $l$, we denote the $s$-th row of $\mathbf{w}^l$ by $\mathbf{w}_s^l$ before reparameterization, and we denote the $s$-th row of $\mathbf{w}_\lambda^l$ by $\mathbf{w}_{\lambda s}^l$ after reparameterization. For simplicity of the notation, we will collect all bias terms in terms $\mathbf{b}, \mathbf{b}_\lambda$ before and after reparameterization respectively. Let

$$F(\mathbf{u}) := \sum_{i=1}^{|S|} \ell(f(x_i; \mathbf{w}^1, \mathbf{w}^2, \ldots, [\mathbf{w}_1^l, \ldots, \mathbf{w}_{s-1}^l, \mathbf{u}, \mathbf{w}_{s+1}^l, \ldots \mathbf{w}_d^l], \ldots, \mathbf{w}^L, \mathbf{b}), y_i)$$

denote the loss as a function on the parameters of the $s$-th neuron in the $l$-th layer (encoded in the $s$-th row of $\mathbf{w}^l$) before reparameterization and

$$\tilde{F}(\mathbf{u}) := \sum_{i=1}^{|S|} \ell(f(x_i; \mathbf{w}_\lambda^1, \mathbf{w}_\lambda^2, \ldots, [\mathbf{w}_{\lambda 1}^l, \ldots, \mathbf{w}_{\lambda(s-1)}^l, \mathbf{u}, \mathbf{w}_{\lambda(s+1)}^l, \ldots \mathbf{w}_{\lambda d}^l], \ldots, \mathbf{w}_\lambda^L, \mathbf{b}_\lambda), y_i)$$

denote the loss as a function on the parameters into the $s$-th neuron in the $l$-th layer (encoded in the $s$-th row of $\mathbf{w}^l$) after reparameterization.

For the same layer $l$, we define a linear function $\eta_s : \mathbb{R}^m \to \mathbb{R}^m$ by

$$\eta_s(\mathbf{u}) = \eta_s(u_1, u_2, \ldots, u_m) = (u_1 \lambda^l_{(s,1)}, u_2 \lambda^l_{(s,2)}, \ldots, u_m \lambda^l_{(s,m)}).$$

By assumption, we have that $\tilde{F}(\eta_s(\mathbf{w}^l_s)) = F(\mathbf{w}^l_s)$ for all $\mathbf{w}^l_s$. By the chain rule, we compute for any coordinate $u_t$ of $\mathbf{u}$,

$$\left. \frac{\partial F(\mathbf{u})}{\partial u_t} \right|_{\mathbf{u}=\mathbf{w}^l_s} = \left. \frac{\partial \tilde{F}(\eta_s(\mathbf{u}))}{\partial u_t} \right|_{\mathbf{u}=\mathbf{w}^l_s}$$

$$= \sum_k \left. \frac{\partial \tilde{F}(\eta_s(\mathbf{u}))}{\partial (\eta_s(\mathbf{u})_k)} \right|_{\eta_s(\mathbf{u})=\eta_s(\mathbf{w}^l_s)} \cdot \left. \frac{\partial (\eta_s(\mathbf{u})_k)}{\partial u_t} \right|_{\eta_s(\mathbf{u})=\eta_s(\mathbf{w}^l_s)}$$

$$= \left. \frac{\partial \tilde{F}(\mathbf{v})}{\partial v_t} \right|_{\mathbf{v}=\mathbf{w}^l_{\lambda s}} \cdot \lambda^l_{(s,t)}.$$

Similarly, for

$$G(\mathbf{u}, \mathbf{u}') := \sum_{i=1}^{|S|} \ell(f(x_i; \mathbf{w}^1, \mathbf{w}^2, \ldots, [\mathbf{w}^l_1, \ldots, \mathbf{w}^l_{s-1}, \mathbf{u}, \mathbf{w}^l_{s+1}, \ldots, \mathbf{w}^l_{s'-1}, \mathbf{u}', \mathbf{w}^l_{s'+1}, \ldots \mathbf{w}^l_d], \ldots$$

$$\ldots, \mathbf{w}^L, , \mathbf{b}), y_i)$$

denoting the loss as a function on the parameters of the $s$-th and $s'$-th neuron in the $l$-th layer (encoded in the $s$-th and $s'$-th row of $\mathbf{w}^l$) before reparameterization and for

$$\tilde{G}(\mathbf{u}, \mathbf{u}') := \sum_{i=1}^{|S|} \ell(f(x_i; \mathbf{w}^1_\lambda, \mathbf{w}^2_\lambda, \ldots$$

$$\ldots, [\mathbf{w}^l_{\lambda 1}, \ldots, \mathbf{w}^l_{\lambda(s-1)}, \mathbf{u}, \mathbf{w}^l_{\lambda(s+1)}, \ldots, \mathbf{w}^l_{s'-1}, \mathbf{u}', \mathbf{w}^l_{s'+1}, \ldots \mathbf{w}^l_{\lambda d}], \ldots$$

$$\ldots \mathbf{w}^L_\lambda, , \mathbf{b}_\lambda), y_i)$$

we have $\tilde{G}(\eta_s(\mathbf{w}^l_s), \eta_{s'}(\mathbf{w}^l_{s'})) = G(\mathbf{w}^l_s, \mathbf{w}^l_{s'})$. For all $s, s', t, t'$ we obtain second derivatives

$$\left. \frac{\partial^2 G(\mathbf{u}, \mathbf{u}')}{\partial u_t \partial u'_{t'}} \right|_{\mathbf{u}=\mathbf{w}^l_s, \mathbf{u}'=\mathbf{w}^l_{s'}} = \lambda^l_{(s,t)} \lambda^l_{(s',t')} \left. \frac{\partial^2 \tilde{G}(\mathbf{u}, \mathbf{u}')}{\partial u_t \partial u'_{t'}} \right|_{\mathbf{u}=\mathbf{w}^l_{\lambda s}, \mathbf{u}'=\mathbf{w}^l_{\lambda s'}}.$$

Consequently, the Hessian $H(\mathbf{w}^l, S)$ of the empirical risk before reparameterization and the Hessian $\tilde{H}(\mathbf{w}^l_\lambda, S)$ after reparameterization satisfy at the position corresponding to $w_{s,t}$ and $w_{s',t'}$ that

$$H_{s,s'}(\mathbf{w}^l, S)_{(t,t')} = \lambda^l_{(s,t)} \lambda^l_{(s',t')} \cdot \tilde{H}_{s,s'}(\mathbf{w}^l_\lambda)_{(t,t')}.$$

Assuming that $\lambda^l_s := \lambda^l_{(s,t)} = \lambda^l_{(s,t')}$ for all $s, t$ and $t'$, then we get that

$$\kappa^l_{Tr}(\mathbf{w}) = \sum_{s,s'=1}^d \langle \mathbf{w}^l_s, \mathbf{w}^l_{s'} \rangle \cdot Tr(H_{s,s'}(\mathbf{w}^l, S))$$

$$= \sum_{s,s'=1}^d \langle \frac{\mathbf{w}^l_{\lambda s}}{\lambda^l_s}, \frac{\mathbf{w}^l_{\lambda s'}}{\lambda^l_{s'}} \rangle \cdot Tr(\lambda^l_s \lambda^l_{s'} \tilde{H}_{s,s'}(\mathbf{w}_\lambda, S))$$

$$= \sum_{s,s'=1}^d \langle \mathbf{w}^l_{\lambda s}, \mathbf{w}^l_{\lambda s'} \rangle \cdot Tr(\tilde{H}_{s,s'}(\mathbf{w}_\lambda, S))$$

$$= \kappa^l_{Tr}(\mathbf{w}_\lambda)$$

This proves Theorem 7.

To show the corollary, we first observe that all layer-wise reparameterizations are covered by the theorem. To see this, we only need to check that the condition $\lambda^l_{(s,t)} = \lambda^l_{(s,t')}$ holds for each $s, t$ and $t'$. For layer-wise reparameterizations, we even have that $\lambda^l_{(s,t)} = \lambda^l$ for all $s, t$, since all weights of one layer are multiplied by the same scalar $\lambda^l$, and $\lambda^l_{(s,t)} = \lambda^l_{(s,t')}$ is easily seen to hold true.

Note further, that any neuron-wise reparameterization given by multiplying all weights into a neuron in a layer $\iota \neq l - 1$ by $\lambda > 0$ and dividing all outgoing weights by $\lambda$ is also covered by the theorem. Hence, the only neuron-wise reparameterization that can change the relative flatness measures is the one multiplying some row of $\mathbf{w}^{l-1}$ by some $\lambda > 0$ and dividing the corresponding column of $\mathbf{w}^l$ by the same $\lambda$. However, by multiplying both $\mathbf{w}^{l-1}$ and $\mathbf{w}^l$ with $V^{-1}$ and $V$ from the left and right respectively, we perform an explicit neuron-wise reparameterization that chooses a unique representative and therefore removes the dependence on such reparamerizations. □

## D.3 Proof of Theorem 5

In this section, we prove Theorem 5. For clarity, we repeat the assumptions and the statement we prove in this section:

We consider a model $f(x, \mathbf{w}) = g(\mathbf{w}\phi(x))$, a loss function $\ell$ and a sample set $S$, and let $O_m \subset \mathbb{R}^{m \times m}$ denote the set of orthogonal matrices. Let $\delta$ be a positive (small) real number and $\mathbf{w} = \omega \in \mathbb{R}^{d \times m}$ denote parameters at a local minimum of the empirical risk on a sample set $S$. If the output function satisfies that $y[\phi_{\delta A}(x_i)] = y[\phi(x_i)] = y_i$ for all $(x_i, y_i) \in S$ and all matrices $||A|| \leq 1$, then we want to show that $f(x, \omega)$ is $((\delta, S, O_m), \epsilon)$-feature robust on average over $O_m$ for $\epsilon = \frac{\delta^2}{2m}\kappa^\phi_{Tr}(\omega) + \mathcal{O}(\delta^3)$, i.e.,

$$\left| \mathcal{E}^\phi_{\mathcal{F}}(f, S, \alpha \mathcal{A}) \right| \leq \frac{\delta^2}{2m}\kappa^\phi_{Tr}(\omega) + \mathcal{O}(\delta^3) \text{ for all } 0 \leq \alpha \leq \delta$$

*Proof.* Writing $z_i = \phi(x_i)$ and $\mathcal{E}_{emp}(\mathbf{w}, S) = \mathcal{E}_{emp}(f(\mathbf{w}, x), S)$ and using the assumption that $y[\phi_{\delta A}(x_i)] = y_i$ for all $(x_i, y_i) \in S$ and all $||A|| \leq 1$, we have for any $0 \leq \alpha \leq \delta$,

$$
\begin{aligned}
\mathcal{E}^\phi_{\mathcal{F}}(f, S, \alpha A) + \mathcal{E}_{emp}(\mathbf{w}, S) &= \frac{1}{|S|} \sum_{i=1}^{|S|} \ell(\psi[\mathbf{w}, \phi_{\alpha A}(x_i)],\ y[\phi_{\alpha A}(x_i)]) \\
&= \frac{1}{|S|} \sum_{i=1}^{|S|} \ell(\psi(\mathbf{w}, z_i + \alpha A z_i), y_i) \\
&= \frac{1}{|S|} \sum_{i=1}^{|S|} \ell(\psi(\mathbf{w} + \alpha \mathbf{w} A, z_i), y_i) \\
&= \mathcal{E}_{emp}(\mathbf{w} + \alpha \mathbf{w} A, S)
\end{aligned}
\tag{14}
$$

The latter is the empirical error $\mathcal{E}_{emp}(\mathbf{w} + \alpha \mathbf{w} A, S)$ of the model $f$ on the sample set $S$ at parameters $\mathbf{w} + \alpha \mathbf{w} A$. If $\delta$ is sufficiently small, then by Taylor expansion around the local minimum $\mathbf{w} = \omega$, we have up to order of $\mathcal{O}(\delta^3)$ that

$$
\begin{aligned}
\mathcal{E}_{emp}(\omega + \alpha \omega A, S) &= \mathcal{E}_{emp}(\omega, S) + \frac{\alpha^2}{2} \sum_{s,t=1}^{d} (\omega_s A) \cdot H_{s,t}(\omega, \phi(S)) \cdot (\omega_t A)^T \\
&\leq \mathcal{E}_{emp}(\omega, S) + \frac{\delta^2}{2} \sum_{s,t=1}^{d} (\omega_s A) \cdot H_{s,t}(\omega, \phi(S)) \cdot (\omega_t A)^T
\end{aligned}
\tag{15}
$$

where $\omega_s$ denotes the $s$-th row of $\omega$.

We consider the set of orthogonal matrices $O_m$ as equipped with the (unique) normalized Haar measure. (For the definition of the Haar measure, see e.g. [17].) We need to show that $\mathbb{E}_{A \sim O_m}\left[ \mathcal{E}^\phi_{\mathcal{F}}(f, S, \alpha A) \right] \leq \frac{\delta^2}{2m} \sum_{s,t} \langle \mathbf{w}_s, \mathbf{w}_t \rangle \cdot Tr(H_{s,t})$ for all $0 \leq \alpha \leq \delta$ with $\mathcal{E}^\phi_{\mathcal{F}}(f, S, A)$ defined as in Eq. 4. Using (14) and (15) we get

$$\mathbb{E}_{A \sim O_m}\left[ \mathcal{E}^\phi_{\mathcal{F}}(f, S, \alpha A) \right] \leq \mathbb{E}_{A \sim O_m}\left[ \frac{\delta^2}{2} \sum_{s,t=1}^{d} (\omega_s A) H_{s,t}(\omega, S)(\omega_t A)^T \right] + \mathcal{O}(\delta^3)$$

Using the unnormalized trace $Tr([m_{s,t}]) = \sum_s m_{s,s}$ we compute with the help of the so-called Hutchinson's trick:

$$
\begin{aligned}
Tr(\mathbb{E}_{A \sim O_m}\left[ (\omega_t A)^T (\omega_s A) \right]) &= \mathbb{E}_{A \sim O_m}\left[ Tr((\omega_t A)^T (\omega_s A)) \right] \\
&= \mathbb{E}_{A \sim O_m}\left[ Tr((\omega_s A)(\omega_t A)^T) \right] \\
&= \mathbb{E}_{A \sim O_m}\left[ Tr(\omega_s \omega_t^T) \right] \\
&= \langle \omega_s, \omega_t \rangle
\end{aligned}
$$

We can interchange two vector coordinates by multiplication of a suitable orthogonal matrix $B$. Since the Haar measure is invariant under multiplication of an orthogonal matrix, the diagonal of $\mathbb{E}_{A \sim O_m} \left[ (\omega_t A)^T (\omega_s A) \right])$ must contain a constant value. This value along the diagonal must then equal $\frac{1}{m} \langle \omega_s, \omega_t \rangle$. Further, we can multiply one vector coordinate by $(-1)$ via multiplication by an orthogonal matrix, and hence the off-diagonal entries of $\mathbb{E}_{A \sim O_m} \left[ (\omega_t A)^T (\omega_s A) \right])$ must be zero, giving that

$$\mathbb{E}_{A \sim O_m} \left[ (\omega_t A)^T (\omega_s A) \right]) = \frac{\langle \omega_s, \omega_t \rangle}{m} \cdot I.$$

Therefore

$$\begin{aligned}
\mathbb{E}_{A \sim O_m} \left[ (\omega_s A) H_{s,t} (\omega_t A)^T \right] &= Tr \left( \mathbb{E}_{A \sim O_m} \left[ (\omega_s A) H_{s,t} (\omega_t A)^T \right] \right) \\
&= \mathbb{E}_{A \sim O_m} \left[ Tr((\omega_s A) H_{s,t} (\omega_t A)^T) \right] \\
&= \mathbb{E}_{A \sim O_m} \left[ Tr(H_{s,t} (\omega_t A)^T (\omega_s A)) \right] \\
&= Tr(H_{s,t} \cdot \mathbb{E}_{A \sim O_m} \left[ (\omega_t A)^T (\omega_s A)) \right] \\
&= Tr(H_{s,t} \cdot \frac{\langle \omega_s, \omega_t \rangle}{m} \cdot I) \\
&= \frac{\langle \omega_s, \omega_t \rangle}{m} Tr(H_{s,t})
\end{aligned}$$

Putting things together, we have for the local optimum $\mathbf{w} = \omega$ that

$$\begin{aligned}
\mathbb{E}_{A \sim O_m} \left[ \mathcal{E}_{\mathcal{F}}^{\phi}(f, S, \alpha A) \right] &\leq \frac{\delta^2}{2} \sum_{s,t=1}^{d} \mathbb{E}_{A \sim O_m} \left[ (\omega_s A) H_{s,t} (\omega_t A)^T \right] + \mathcal{O}(\delta^3) \\
&= \frac{\delta^2}{2m} \sum_{s,t=1}^{d} \langle \omega_s, \omega_t \rangle \cdot Tr(H_{s,t}) + \mathcal{O}(\delta^3) \\
&= \frac{\delta^2}{2m} \kappa_{Tr}^{\phi}(\omega) + \mathcal{O}(\delta^3)
\end{aligned}$$

$\square$

We can further generalize Theorem 5 to more complex labels by introducing a notion of approximately locally constant labels. The following definition frees us from the strong assumption of locally constant labels, i.e. $y[\phi_{\delta A}(x_i)] = y[\phi(x_i)] = y_i$ for all $(x_i, y_i) \in S$ and all matrices $||A|| \leq 1$, while still restricting label changes to be one order smaller than the contribution of flatness.

**Definition 9.** *Let $\mathcal{D}$ be a data distribution on a labeled sample space $\mathcal{X} \times \mathcal{Y}$ and $S$ a finite iid sample of $\mathcal{D}$. Let $f = \psi \circ \phi$ be a model composed into a feature extractor $\phi$ and predictor $\psi$. We say that $\mathcal{D}$ has approximately locally constant labels of order three around the points $(x, y) \in S$ in feature space $\phi$, if there is some constant $C$ such that*

$$\frac{1}{|S|} \sum_{i=1}^{|S|} \left| \ell(\psi(\phi(x_i) + \Delta_i), y[\phi(x_i) + \Delta_i]) - \ell(\psi(\phi(x_i) + \Delta_i), y_i) \right| \leq C\delta^3 \text{ for } ||\Delta_i|| \leq \delta ||\phi(x_i)||$$

**Corollary 10.** *Consider a model $f(x, \mathbf{w}) = \psi(\mathbf{w}, \phi(x)) = g(\mathbf{w}\phi(x))$ as above, a loss function $\ell$ and a sample set $S$, and let $O_m \subset \mathbb{R}^{m \times m}$ denote the set of orthogonal matrices. Let $\delta$ be a positive (small) real number and $\mathbf{w} = \omega \in \mathbb{R}^{d \times m}$ denote parameters at a local minimum of the empirical risk on a sample set $S$. If $\mathcal{D}$ has approximately locally constant labels of order three around the points $(x, y) \in S$ in feature space, then $f(x, \omega)$ is $((\delta, S, O_m), \epsilon)$-feature robust on average over $O_m$ for $\epsilon = \frac{\delta^2}{2m} \kappa_{Tr}^{\phi}(\omega) + \mathcal{O}(\delta^3)$.*

*Proof.* As before, we abbreviate $\phi(x_i)$ by $z_i$. We only need to modify (14) to account for the strictly weaker assumption on the labels. For this, we perform Taylor approximation with respect to the

labels at $y[\phi(x_i)] = y_i$ to obtain

$$
\begin{aligned}
\mathcal{E}_{\mathcal{F}}^{\phi}(f, S, \alpha A) + \mathcal{E}_{emp}(\mathbf{w}, S) &= \frac{1}{|S|} \sum_{i=1}^{|S|} \ell(\psi[\phi_{\alpha A}(x_i)], \ y[\phi_{\alpha A}(x_i)]) \\
&\overset{Def\ 9}{\leq} \frac{1}{|S|} \sum_{i=1}^{|S|} \ell(\psi(\mathbf{w}, z_i + \alpha A z_i), y_i) + \mathcal{O}(\delta^3) \\
&= \frac{1}{|S|} \sum_{i=1}^{|S|} \ell(\psi(\mathbf{w} + \alpha \mathbf{w} A, z_i), y_i) + \mathcal{O}(\delta^3) \\
&= \mathcal{E}_{emp}(\mathbf{w} + \alpha \mathbf{w} A, S) + \mathcal{O}(\delta^3)
\end{aligned}
$$

The rest of the proof follows the arguments used to show Theorem 5.

$\square$

## D.4 Proof of Theorem 6

To prove Theorem 6, we will require a proposition that bounds $\epsilon$-representativeness for $\lambda_i$ the local densities from Proposition 2. This is achieved in Proposition 12 below uniformly over all distributions $\mathcal{D}$ that satisfy mild regularity assumptions necessary for a well-defined kernel density estimation. We first compose the proof to Theorem 6 and subsequently show the arguments leading to the required proposition.

The main idea to prove Theorem 6 is that the family of distributions considered in Proposition 2 (a) provides an explicit link between $\epsilon$-representativeness and feature robustness (Proposition 2 and Equation 5), (b) allows us to approximately bound feature robustness by relative flatness (Theorem 5), and (c) allows us to apply a kernel density estimation to uniformly bound $\epsilon$-representativeness (Proposition 12).

Theorem 6 is the informal counterpart to the following version.

**Theorem 11.** *Consider a model $f(x, \mathbf{w}) = g(\mathbf{w}\phi(x))$, a loss function $\ell$, a sample set $S$, and let $m$ denote the dimension of the feature space defined by $\phi$ and let $\delta$ be a positive (small) real number. Let $\omega \in \mathbb{R}^{d \times m}$ denote a local minimum of the empirical risk on an iid sample set $S$.*

*Suppose that the distribution $\mathcal{D}$ has a smooth density $p_{\mathcal{D}}^\phi$ on the feature space $\mathbb{R}^m$ such that $\int_z \left| \nabla^2 \left( p_{\mathcal{D}}^\phi(z) \|z\|^2 \right) \right| dz$ and $\int_z \frac{p_{\mathcal{D}}^\phi(z)}{\|z\|^m} dz$ are well-defined and finite. Then for sufficiently large sample size $|S|$, if the distribution has approximately locally constant labels of order three (see Definition 9), then it holds with probability $1 - \Delta$ over sample sets $S$ that*

$$\mathcal{E}_{gen}(f(\cdot, \omega), S) \lesssim |S|^{-\frac{2}{4+m}} \left( \frac{\kappa_{Tr}^\phi(\omega)}{2m} + C_1(p_{\mathcal{D}}^\phi, L) + \frac{C_2(p_{\mathcal{D}}^\phi, L)}{\sqrt{\Delta}} \right)$$

*up to higher orders in $|S|^{-1}$ for constants $C_1, C_2$ that depend only on the distribution in feature space $p_{\mathcal{D}}^\phi$ induced by $\phi$ and the chosen $|S|$-tuple $\Lambda_\delta$ as in Proposition 2 and the maximal loss $L$.*

*Proof.* The proof combines Equation 5 with Proposition 12 and Theorem 5. At first we use Equation 5 to split the generalization gap into $\mathcal{E}_{gen}(f) = \mathcal{E}_{Rep}^\phi(f, S, \Lambda_{\mathcal{A}_\delta}) + \mathcal{E}_{\mathcal{F}}(f, S, \mathcal{A}_\delta)$. For the family of distributions $\Lambda_\delta$ from Proposition 2, we have by Proposition 12 that

$$|\mathcal{E}_{Rep}^\phi(\psi \circ \phi, S, \Lambda_\delta^\phi)| \leq \left( C_1(p_{\mathcal{D}}^\phi, L) + \frac{C_2(p_{\mathcal{D}}^\phi, L)}{\sqrt{\Delta}} \right) \cdot |S|^{-\frac{2}{4+m}} + \mathcal{O}(|S|^{-\frac{3}{4+m}})$$

when $\delta = |S|^{-\frac{1}{4+m}}$. With $\mathcal{E}_{\mathcal{F}}(f, S, \mathcal{A}_\delta) = \mathbb{E}_{A \sim \mathcal{A}_\delta} \left[ \mathcal{E}_{\mathcal{F}}(f, S, A) \right]$, we use (the proof to) Proposition 2 to see that this can be written as

$$\mathbb{E}_{A \sim \mathcal{A}_\delta} \left[ \mathcal{E}_{\mathcal{F}}(f, S, A) \right] = \mathbb{E}_{0 \leq \alpha \leq \delta} \left[ \mathbb{E}_{A \sim (\mathcal{O}_m, \mu)} \left[ \mathcal{E}_{\mathcal{F}}(f, S, \alpha A) \right] \right]$$

where $O_m \subset \mathbb{R}^{m \times m}$ denote the set of orthogonal matrices and $\mu$ the Haar measure on this set. Finally, Theorem 5 bounds the latter by $\frac{|S|^{-\frac{2}{4+m}}}{2m} \kappa_{Tr}^\phi(\omega)$ up to higher orders in $|S|^{-1}$. $\square$

We finally prove that the bound on $\epsilon$-representativeness in the proof to the preceding Theorem indeed holds true.

**Proposition 12.** *Consider a model $f(x, \mathbf{w}) = \psi(\mathbf{w}, \phi(x))$, a loss function $\ell$ and let $S \subseteq \mathcal{X} \times \mathcal{Y}$ be a finite sample set. With $x_i \in S$, let $\lambda_i(z) = k_{\delta\|\phi(x_i)\|}(0, z)$ define an $|S|$-tuple $\Lambda_\delta$ of densities as in Proposition 2 and assume that the loss function is bounded by $L$. Suppose that the distribution $\mathcal{D}$ has a smooth density $p_{\mathcal{D}}^\phi$ on a feature space $\mathbb{R}^m$ such that $\int_z \nabla^2 \left( p_{\mathcal{D}}^\phi(z) \|z\|^2 \right) dz$ and $\int_z \frac{p_{\mathcal{D}}^\phi(z)}{\|z\|^m} dz$ are well-defined and finite. Then there exist constants $C_1(p_{\mathcal{D}}^\phi, L), C_2(p_{\mathcal{D}}^\phi, L)$ depending on the distribution and the maximal loss such that, with probability $1 - \Delta$ over possible sample sets $S$, $\epsilon$-interpolation is bounded for $\delta = |S|^{-\frac{1}{4+m}}$ by*

$$|\mathcal{E}_{Rep}^\phi(f, S, \Lambda_\delta)| \leq \left( C_1(p_{\mathcal{D}}^\phi, L) + \frac{C_2(p_{\mathcal{D}}^\phi, L)}{\sqrt{\Delta}} \right) \cdot |S|^{-\frac{2}{4+m}} + \mathcal{O}(|S|^{-\frac{3}{4+m}})$$

*Proof.* We let

$$\hat{p}(z) = \frac{1}{|S|} \sum_{i=1}^{|S|} k_{\delta||\phi(x_i)||}(\phi(x_i), z)$$

With $\lambda_i = k_{\delta||\phi(x_i)||}(0, z)$ we have

$$\left| \mathcal{E}^{\phi}_{Rep}(f, S, \Lambda_\delta) \right| = \left| \mathcal{E}(f) - \frac{1}{|S|} \sum_{i=1}^{|S|} \mathbb{E}_{\xi \sim \lambda_i} \left[ \ell(\psi(\phi(x_i) + \xi), y[\phi(x_i) + \xi]) \right] \right|$$

$$= \left| \int_z p^{\phi}_{\mathcal{D}}(z) \cdot \ell(\psi(z), y(z)) \, dz - \frac{1}{|S|} \sum_{i=1}^{|S|} \int_z k_{\delta||\phi(x_i)||}(\phi(x_i), z) \cdot \ell(\psi(z), y(z)) \, dz \right| \quad (16)$$

$$\leq \underbrace{\left| \int_z (p^{\phi}_{\mathcal{D}}(z) - \mathbb{E}_S[\hat{p}(z)]) \cdot \ell(\psi(z), y(z)) \, dz \right|}_{(I)} + \underbrace{\left| \int_z (\mathbb{E}_S[\hat{p}(z)] - \hat{p}(z)) \cdot \ell(\psi(z), y(z)) \, dz \right|}_{(II)}$$

For the further analysis, we make use of Jones et al. [15] and combine it with the generalization to the multivariate case in Chp. 4.3.1 in Silverman [33]. A Taylor approximation with respect to the bandwidth of the kernel $\delta$ yields

$$(I) = \frac{\delta^2}{2} \tau_2 \left| \int_z \nabla^2 \left( p^{\phi}_{\mathcal{D}}(z)||z||^2 \right) \ell(\psi(z), y(z)) dz \right| + \mathcal{O}(\delta^3)$$

where

$$\tau_2 = \int_z ||z||^2 k_1(0, z) dz.$$

For (II) we consider the random variable $Z = \int_z \hat{p}(z) \ell(\psi(z), y(z)) \, dz$ as a function on the set of possible sample sets of a fixed size. Applying Chebychef's inequality on $Z$, we get that

$$Pr\left( \left| Z - \mathbb{E}_S\left[ Z \right] \right| > \epsilon_{est} \right) \leq \frac{Var(Z)}{\epsilon_{est}^2} =: \Delta \ .$$

Solving for $\epsilon_{est}$ yields that with probability $1 - \Delta$ we have

$$(II) = |Z - \mathbb{E}_S\left[ Z \right]| \leq \frac{\sqrt{Var(Z)}}{\sqrt{\Delta}}$$

Further, the variance of $Z$ can be bounded by

$$Var(Z) = \mathbb{E}_S\left[ (Z - \mathbb{E}_S\left[ Z \right])^2 \right]$$

$$= \mathbb{E}_S\left[ \left( \int \hat{p}(z) \ell(\psi(z), y(z)) \, dz - \mathbb{E}_S\left[ \int \hat{p}(z) \ell(\psi(z), y(z)) \, dz \right] \right)^2 \right]$$

$$= \mathbb{E}_S\left[ \left( \int \left( \hat{p}(z) - \mathbb{E}_S\left[ \hat{p}(z) \right] \right) \ell(\psi(z), y(z)) \, dz \right)^2 \right]$$

$$\leq \underbrace{\mathbb{E}_S\left[ \int \left( \hat{p}(z) - \mathbb{E}_S\left[ \int \hat{p}(z) \right] \right)^2 dz \right]}_{(III)} \cdot \underbrace{\left( \int_z \ell(\psi(z), y(z))^2 \, dz \right)}_{\leq L^2 \text{Vol}(\phi(\mathcal{D}))}$$

It follows from Eq. (2.3) in Jones et al. [15] together with Eq. 4.10 in Silverman [33] for (III) that for small $\delta$ and large sample size $|S|$ the term (III), i.e., the variance of $\tilde{p}$, is given by

$$(III) = \beta |S|^{-1} \delta^{-m} \alpha + \mathcal{O}(|S|^{-2}) \ ,$$

where $\alpha = \int_z \frac{p^{\phi}_{\mathcal{D}}(z)}{||z||^m} \, dz$ and $\beta = \int_z k_1(0, z)^2 \, dz$. Putting things together gives

$$\mathcal{E}^{\phi}_{Rep}(f, S, \Lambda_\delta)| \leq L \frac{\delta^2}{2} \tau_2 \left| \int_z \nabla^2 \left( p^{\phi}_{\mathcal{D}}(z)||z||^2 \right) \right| dz + \frac{L\sqrt{\alpha\beta}}{\sqrt{\Delta}} \sqrt{\text{Vol}(\phi(\mathcal{D}))} |S|^{-\frac{1}{2}} \delta^{-\frac{m}{2}}$$

$$+ \mathcal{O}(|S|^{-2}) + \mathcal{O}(\delta^3) \ .$$

Choosing the bandwidth as $\delta = |S|^{-\frac{1}{4+m}}$ gives

$$|\mathcal{E}_{Rep}^{\phi}(f, S, \Lambda_{\delta})| \leq |S|^{-\frac{2}{4+m}} \left( \tau_2 L \left| \int_z \nabla^2 \left( p_{\mathcal{D}}^{\phi}(z) ||z||^2 \right) \right| dz + \frac{\sqrt{\alpha\beta}L}{\sqrt{\Delta}} \sqrt{\text{Vol}(\phi(\mathcal{D}))} \right)$$
$$+ \mathcal{O}(|S|^{-\frac{3}{m+4}}) .$$

The result follows from setting

$$C_1 = \tau_2 L \left| \int_z \nabla^2 \left( p_{\mathcal{D}}^{\phi}(z) ||z||^2 \right) \right| dz$$
$$C_2 = \sqrt{\alpha\beta} L \sqrt{\text{Vol}(\phi(\mathcal{D}))} .$$

$\square$

# E  Relative flatness for a uniform bound over general distributions on feature matrices

This article based its consideration on the specific distribution on feature matrices of Proposition 2, since this distribution allows to use standard results of kernel density estimation in the proof to Theorem 6. However, the decomposition of the risk in Equation 5 holds for any distribution on feature matrices $\mathcal{A}$ and induced distributions on feature space $\Lambda_{\mathcal{A}}$. To allow maximal flexibility in the choice of a distribution $\mathcal{A}$ on feature matrices of norm $||A|| \leq 1$, we define another version of relative flatness based on the maximal eigenvalues of partial Hessians instead of the trace.

**Definition 13.** *For a model $f(\mathbf{w}, x) = g(\mathbf{w}\phi(x))$ with a twice differentiable function $g$, a twice differentiable loss function $\ell$ and a sample set $S$ we define maximal relative flatness by*

$$\kappa^{\phi}(\mathbf{w}) := \sum_{s=1}^{d} ||\mathbf{w}_s||^2 \cdot \lambda_{max}(H_{s,s}(\mathbf{w}, \phi(S))) \tag{17}$$

*where $\lambda_{max}$ denotes the maximal eigenvalue of a matrix and $H_{s,s'}$ the Hessian matrix as in (8).*

The analogue to Theorem 5 for maximal relative flatness shows that maximal flatness bounds feature robustness uniformly over all feature matrices of norm $||A|| \leq 1$.

**Theorem 14.** *Consider a model $f(x, \mathbf{w}) = g(\mathbf{w}\phi(x))$ as above, a loss function $\ell$ and a sample set $S$, and let $O_m \subset \mathbb{R}^{m \times m}$ denote the set of orthogonal matrices. Let $\delta$ be a positive (small) real number and $\mathbf{w} = \omega \in \mathbb{R}^{d \times m}$ denote parameters at a local minimum of the empirical risk on a sample set $S$. If the labels satisfy that $y[\phi_{\delta A}(x_i)] = y[\phi(x_i)] = y_i$ for all $(x_i, y_i) \in S$ and all $||A|| \leq 1$, then, for each feature selection matrix $||A|| \leq 1$ the model $f(x, \omega)$ is $((\delta, S, A), \epsilon)$-feature robust for $\epsilon = \frac{\delta^2 d}{2} \kappa^{\phi}(\omega) + \mathcal{O}(\delta^3)$*

*Proof.* Writing $z_i = \phi(x_i)$ and $\mathcal{E}_{emp}(\mathbf{w}, S) = \mathcal{E}_{emp}(f(\mathbf{w}, x), S)$ and using the assumption that $y[\phi_{\delta A}(x_i)] = y_i$ for all $(x_i, y_i) \in S$ and all $||A|| \leq 1$, we have by the first part of the proof of Theorem 5 that for any $0 \leq \alpha \leq \delta$,

$$\mathcal{E}_{\mathcal{F}}^{\phi}(f, S, \alpha A) + \mathcal{E}_{emp}(\mathbf{w}, S) = \mathcal{E}_{emp}(\mathbf{w} + \alpha \mathbf{w} A, S) \tag{18}$$

and

$$\mathcal{E}_{emp}(\omega + \alpha \omega A, S) \leq \mathcal{E}_{emp}(\omega, S) + \frac{\delta^2}{2} \sum_{s,t=1}^{d} (\omega_s A) \cdot H_{s,t}(\omega, \phi(S)) \cdot (\omega_t A)^T + \mathcal{O}(\delta^3) \tag{19}$$

at a local minimum $\omega$, where $\omega_s$ denotes the $s$-th row of $\omega$.

Note that for $||A|| \leq 1$ and a row vectors $\mathbf{w}_s$ it holds that $||\mathbf{w}_s A|| \leq ||\mathbf{w}_s||$. Further, since the full Hessian matrix $H(\omega, S) = (H_{s,t}(\omega, S))_{s,t}$ is a positive semidefinite matrix at a local minimum $\omega$, it holds for each row vectors $\mathbf{w}_s, \mathbf{w}_t$ that

$$\mathbf{w}_s H_{s,t}(\omega, S) \mathbf{w}_t^T \leq \frac{1}{2} \Big( \mathbf{w}_s H_{s,s}(\omega, S) \mathbf{w}_s^T + \mathbf{w}_t H_{t,t}(\omega, S) \mathbf{w}_t^T \Big), \tag{20}$$

We therefore get that for any feature matrix $A$ with $||A|| \leq 1$,

$$\mathcal{E}_{\mathcal{F}}^{\phi}(f, S, \delta A) \leq \max_{||A|| \leq 1} \mathcal{E}_{\mathcal{F}}(f, S, \delta A)$$

$$\stackrel{(18),(19)}{\leq} \max_{||A|| \leq 1} \frac{\delta^2}{2} \sum_{s,t=1}^{d} (\omega_s A) \cdot H_{s,t}(\omega, S) \cdot (\omega_t A)^T + \mathcal{O}(\delta^3)$$

$$\stackrel{(20)}{\leq} \max_{||A|| \leq 1} \frac{\delta^2 d}{2} \sum_{s=1}^{d} (\omega_s A) \cdot H_{s,s}(\omega, S) \cdot (\omega_s A)^T + \mathcal{O}(\delta^3)$$

$$\leq \frac{\delta^2 d}{2} \sum_{s=1}^{d} \max_{||\mathbf{z}|| \leq ||\omega_s||} \mathbf{z} H_{s,s}(\omega, S) \mathbf{z}^T + \mathcal{O}(\delta^3) \qquad (21)$$

$$= \frac{\delta^2 d}{2} \sum_{s=1}^{d} \max_{||\mathbf{z}||=1} ||\omega_s||^2 \, \mathbf{z} H_{s,s}(\omega, S) \mathbf{z}^T + \mathcal{O}(\delta^3)$$

$$= \frac{\delta^2 d}{2} \sum_{s=1}^{d} ||\omega_s||^2 \, \lambda_{max}(H_{s,s}(\omega, S)) + \mathcal{O}(\delta^3)$$

$$= \frac{\delta^2 d}{2} \kappa^{\phi}(\omega) + \mathcal{O}(\delta^3)$$

where we used the identity that $\max_{||x||=1} x^T M x = \lambda_{max}(M)$ for any symmetric matrix $M$.

$\square$

With this, the analogue to Theorem 6 (or its version Theorem 11 in the appendix) allows maximal flexibility to choose $\mathcal{A}_{\delta}$ (and $\delta > 0$) to bound representativeness. This leads to the following generalization bound.

**Theorem 15.** *Consider a model $f(x, \mathbf{w}) = g(\mathbf{w}\phi(x))$, a loss function $\ell$, a sample set $S$, and let $m$ denote the dimension of the feature space defined by $\phi$ and let $\delta$ be a positive (small) real number. Let $\omega \in \mathbb{R}^{d \times m}$ denote a local minimum of the empirical risk on an iid sample set $S$.*

*Let $\Upsilon_{\delta}$ be the set of all $|S|$-tuple of distributions $\Lambda_{\mathcal{A}_{\delta}}$ on feature vectors induced by a distribution $\mathcal{A}_{\delta}$ on feature matrices of norm smaller than $\delta$ as in Section 3. Then it holds that*

$$\mathcal{E}_{gen}(f(\cdot, \omega), S) \leq \inf_{\mathcal{A}_{\delta} \in \Upsilon_{\delta}} \mathcal{E}_{Rep}^{\phi}(f, S, \Lambda_{\mathcal{A}_{\delta}}) + \frac{\delta^2 d}{2} \sum_{s=1}^{d} \kappa^{\phi}(\omega) + \mathcal{O}(\delta^3).$$

*Proof.* Part (i) follows from combining Equation 5 with Theorem 14. First, we use (5) to split the generalization gap into $\mathcal{E}_{gen}(f) = \mathcal{E}_{Rep}^{\phi}(f, S, \Lambda_{\mathcal{A}_{\delta}}) + \mathcal{E}_{\mathcal{F}}(f, S, \mathcal{A}_{\delta})$. Then, Theorem 14 shows that $\mathcal{E}_{\mathcal{F}}(f, S, \mathcal{A}_{\delta}) \leq \frac{\delta^2 d}{2} \kappa^{\phi}(\omega) + \mathcal{O}(\delta^3)$ as $\mathcal{E}_{\mathcal{F}}(f, S, \delta A) \leq \frac{\delta^2 d}{2} \kappa^{\phi}(\omega) + \mathcal{O}(\delta^3)$ for all $||A|| \leq 1$. $\square$