# OpenReview forum: "Relative Flatness and Generalization"
_NeurIPS.cc/2021/Conference — NeurIPS 2021 Poster_

### Official Review · Reviewer_bRT2 · 2021-07-19

**Rating:** 6
**Confidence:** 3

**Summary:**

This paper develops a theoretical approach to tackling the relationship between flatness of the loss landscape around trained models and generalization. To accomplish this, the authors need to make certain assumptions about the data such as locally constant labels as well as its representativeness. The authors show that flatness with respect to the last layer parameters is related to feature robustness, and ultimately show that generalization performance can be bounded by their flatness quantity. The paper then presents a few experiments on synthetic datasets or small models to test the performance of the theory. They find that their measure performs similarly to others such as PacBayes and Trace.

**Limitations And Societal Impact:**

The authors discussed the limitations with respect to approximating representativeness, but a discussion of other limitations would be good. There is not a societal impacts section but I don’t know if that’s necessary for this paper.

**Main Review:**

Unless I misunderstood (which is definitely possible!) the theory only applies to models with a fixed feature embedding and where only the last layer parameters are trainable. If so, this limits the general applicability of the work and my excitement for it. It would be good for the authors to comment one way or another on this, perhaps in a limitations section which is missing. (Or not if I misunderstood.)

What is the measure of correlation shown in Figure 5? Is it rank correlation? More details on the experimental setup would be appreciated.

My interpretation is that their measure works about as well as the other well-performing measures in the plot. Could the authors comment if the improvement should be considered significant?

In the relative flatness and trace measures in Figure 5, it looks like they both have a vertical “clump” of points that behaves very differently from the other points. Do the authors understand the origin of this clump? Is there a particular set of hyper parameters for which the measure does not seem to change? In some sense, this is worse performance to me than say PacBayes, which doesn’t seem to exhibit this.



**Time Spent Reviewing:**

3

---

> ### Author Response · Authors · 2021-08-10
> **Response to Reviewer bRT2**
>
> We thank the reviewer for the overall positive evaluation of our work and the suggestions for further explanations of the extent of our contribution. Indeed, there seems to be a misunderstanding about the setup and generality of our theory which we want to clarify. We apologize if that was not sufficiently clear from the paper.
>
> **Does the theory only apply to the last layer?**
>
> Our theory does not only apply to models with a fixed feature embedding and where only the last layer parameters are trainable, but our theory applies to all models that can be expressed as a feature extractor and model as $f(x) =\psi(w,\phi(x))= g(w \phi(x))$, “which includes linear and kernel models, as well as most neural networks” [line 41]. In particular, our experiments on large networks and a non-trivial dataset [reviewer zbyb] are the outcome of training the entire network (ResNET 18) and not only the last layer.  There is also no restriction on the choice of layer of a neural network to consider as feature representation. A natural and convenient choice is the last layer (mostly because it is usually less wide and thus the computation of the Hessian is less time consuming), but our theory applies to any layer. To support this empirically, we ran additional experiments on MNIST that show the strong correlation of relative flatness with generalization holds for all hidden layers (please see the reply to reviewer 5qca for further details). Since our theory evaluates the generalization gap for trained models, we consider both $\phi$ and $\psi$ to be given for each individual model in question, i.e., we do not consider retraining or fine-tuning $\psi$ for fixed $\phi$. It is of course an interesting question whether models can be trained directly to achieve high relative flatness (e.g., with a suitable regularization term), but this goes beyond the scope of this work.
>
> **Is the improvement of the relative flatness measure significant?**
>
> As all reviewers agree, the main contribution of the paper is the non-trivial and  relevant theoretical connection between flatness - a local property - and generalization. The relative flatness measure is a natural outcome of our theoretical analysis.
>
> Using the trace of the Hessian in relative flatness gives the most straight-forward variant of relative flatness which is already a significant improvement, outperforming all competing measures on CIFAR10 for 110 trained deep neural networks. While the improvement over PAC Bayes is small in Figure 5 (Pearson correlation 0.81 vs 0.79), the PAC Bayes measure is not reparameterization invariant. That is, the same network function can result in different values for the PAC Bayes flatness measure. Figure 7 in the supplements presents results with reparamaterizations of the network, where our flatness measure still shows a correlation factor of 0.81 (due to its reparameterization-independence), but PAC Bayes declines to a correlation factor of 0.27. (It is conceivable that the reparameterized functions can also be achieved by training using different initialization techniques instead of manual reparameterizations, so these results are not simply a theoretical construct.)
>
> The relative flatness measure can be refined by using a (different) weighted sum of the eigenvalues of the Hessian (see also our reply to 5qca), instead of the trace of the Hessian. This weighted sum has to be adapted to the underlying data distribution, the learned feature representation, and the choice of a suitable distribution on matrices. Thus, there is great potential for further improvement.
>
> **Can you explain the vertical clump of points in Fig. 5?**
>
> Most points in this vertical clump are networks trained with SGD, whereas the other points are mostly the ones trained with RMSprop and Adam. This supports the findings in our references [14,48] that SGD converges to flatter minima than other optimization techniques.
> The PAC-Bayes measure has a stronger dependence on weight norm, thus showing a larger spread, and has shown good performance in our experiments as well as in the large-scale comparative study of Jiang et al. [15] (where it was outperformed by a robustness-based sharpness score - in spirit similar to our feature robustness). The major downside of the PAC-Bayes measure is that it is not reparameterization-invariant.
>
> **What is the measure of correlation shown in Fig 5?**
>
> We thank the reviewer for pointing out the missing detail. We use Pearson correlation in the experiments. Moreover, we considered the last hidden layer. We will add this to the experiments sections. We would be thankful if the reviewer could direct us to additional experimental details we may have missed to describe.
>
> **Experiments that support the theory.**
>
> The results for the relative flatness measure were obtained on a non-trivial dataset with a pretty large network [reviewer zbyb]. In addition, we provided synthetic experiments on a smaller MLP architecture. We performed additional experiments for this discussion: an analysis of the impact of the choice of reparameterization (i.e., choice of layer) on the correlation between the relative flatness measure and generalization on MNIST [see the reply to reviewer 5qca], and two additional synthetic experiments on approximating the generalization bound, one for a deeper network and one for a more complex dataset [see the reply to reviewer zbyb]. While the paper is theoretical by nature, we hope that this range of experiments convincingly supports the theoretical results.

---

> > ### Comment · Reviewer_bRT2 · 2021-09-01
> > **Thanks for the response and clarifications**
> >
> > I've updated my rating from 5 to 6.
> >
> > A weighted sum of Hessian eigenvalues would indeed be an interesting direction, though that seems like a somewhat different direction from the current work. I'm curious the authors' thoughts on whether lack of reparameterization invariance being treated fairly (not just by them!) since there is a special parameterization, namely the one used to train the model.

---

### Official Review · Reviewer_zbyb · 2021-07-20

**Rating:** 7
**Confidence:** 2

**Summary:**

The paper investigates the connection between loss-based flatness and generalization. The authors relate flatness to interpolation from representative data in the feature space, and use two notions to do that: 1) representativness and 2) feature robustness. They derive a new flatness metric -- a relative flatness -- and on a small set of empirical observations show that it correlates well with generalization and that it has some other desirable theoretical properties such as reparametrization invariance.

**Limitations And Societal Impact:**

Yes

**Main Review:**

I like the paper, it deals with and important problem, is pretty clearly written, and contains both theory and an empirical validation. Here are some of my detailed points, questions, and potential problems that could be addressed by the authors to make the paper better:

THE POSITIVES:
1) the paper deals with an important question of the relationship between local loss Hessian based flatness and generalization. Flatness is a local property of the loss landscape, while generalization is a much higher level concept. A lot has been written on the connection, but as far as I can tell there hasn't been a satisfactory explanation for why the connection exists. This paper is a good step in that direction.
2) The paper is pretty clearly written and I had an easy time following the argument.
3) I appreciate that the authors ran an empirical validation of their theoretical results using a pretty large network and a non-trivial dataset (CIFAR-10) rather than just synthetic data.
4) I like that the authors discuss the limitations of their approach, such as the assumption of constant labels.

QUESTIONS AND WEAKER POINTS:
1) Constant labels. I am a bit confused about the constant labels assumption. If I get it correctly, the assumption is that when you look at the partitioning of the input / feature space, as discussed from line 90 further, the cells are considered to have the same true label as the centroid of the cell? This seems pretty reasonable on the face of it, but I do wonder if small soft changes in the true label, e.g. from (1,0,0) to (1-e,e/2,e/2) as you move within the cell would do something to your theory? I think it is entirely plausible that the labels could be changing as judged by a human as a reference for the ground truth. Furthermore, how does the reference 35 use the constant labels? I didn't get that connection.

2) The partitioning you are looking at could be e.g. a Voronoi partitioning (a minor point)? Do distances matter. Do shapes of the cells make any difference. Do they have to be convex? ​

3) The verification in section 6 is only done on a single architecture and dataset, yet the validity of the measure is asserted generally. I think it would be better to do a broader set of experiments, including other data modalities and networks, given the breath of your theoretical claim. I am not suggesting you need to go to imagenet directly, but more diversity in architectures and dataset would be good to add.

RELATED PAPERS?
I remember seeing a proof relating the changes in weights to changes in features / logits in https://arxiv.org/abs/2002.03432 which might be relevant to you. Another paper that I got reminded of in your summary of the works on Tr(H) was https://arxiv.org/abs/1807.02581 .

VERY  MINOR STUFF:
1) I think  \mathrm{Tr} for an operator looks better than Tr
2) the colors in Figure 5 are hard to distinguish, could you use more different colors?

UPDATE AFTER REPLY:
I am happy with the detailed response and increase my score from 6 to 7. Thank you!



**Time Spent Reviewing:**

3

---

> ### Author Response · Authors · 2021-08-10
> **Response to Reviewer zbyb**
>
> We thank the reviewer for the careful and encouraging evaluation, in particular for pointing out that the paper is a good step towards a satisfactory explanation for the non-trivial connection between flatness and generalization, and that the paper supports the theoretical analysis with experiments on large networks and a non-trivial dataset.
>
> **Does the theory also apply for soft label changes?**
>
> The theory also applies to soft label changes and given a certain bound on the change rate, the results are similar. For simplicity, we presented the results for “hard” constant labels in the main text. A generalization of Theorem 5 to soft label changes is provided in the appendix (line 864--874 in appendix, Def 9 and Corollary 10).
>
> **Does distance or shape of neighborhoods in which labels are constant matter?**
>
> While we start from a true partitioning of the input space (where a Voronoi partition could be used, but may not be a theoretically ideal choice), the approximating notion of eps-representativeness is  closer to a cover of the input distribution using shapes around the training points. The shapes and distances are defined by the $\lambda_i$ (respectively the distributions on matrices A) and the parameter $\delta$. The size of the shape around a training point is then determined by $\delta$ times the norm of the training point, since we use multiplicative noise (using additive noise is possible, but would lead to a different flatness measure with less attractive properties such as reparameterization-independence). The shapes can be arbitrarily complex (including non-convex shapes). Ideally, the shapes that lead to an optimal coverage of the input distribution (i.e., small eps-representativeness) match the shapes induced by the distribution on the matrices A from small feature robustness. This will lead to a very tight generalization bound. For the relative flatness measure, the shapes are given by the kernel used to bound feature robustness, i.e., truncated Gaussians. We add a more detailed discussion on the relationship between the shapes in the cover of the input distribution, the distributions over A for feature robustness, and relative flatness to the final version of the manuscript.
>
> **How does reference 35 use constant labels?**
>
> An adversarial example is defined as an “imperceptible non-random” [35] local perturbation within a small radius around an example that changes the prediction. Here, imperceptible implies that the true class of the example should not change, i.e., labels are locally constant. The paper clarifies this by stating that in deep learning often an implicit local smoothness assumption is made: “In general, imperceptibly tiny perturbations of a given image do not normally change the underlying class” [35], and that this assumption is “typically valid for computer vision problems” [35].
>
> **Broader set of experiments, including more data modalities and networks.**
>
> As the reviewer pointed out, we performed the verification of the relative flatness measure on a large network and a non-trivial dataset. The synthetic experiments were done on smaller MLPs with controlled feature representation to showcase the validity of the theory. We agree with the reviewer that confirming these results on other architectures and data modalities adds further support to the theory. Thus, we conducted the experiment on approximating representativeness on a larger architecture (784 x 512 x 384 x 256 x 196 x 128 x 32 x 10 MLP, in contrast to the 784 x 512 x 128 x 16 x 10 MLP used in the original experiment). The results (see https://anonymous.4open.science/r/RelativeFlatnessAndGeneralization-B175/AdditionalExperiments/genBoundApproxDeep.pdf) are similar. Moreover, we used a more complex synthetic dataset. Originally, we sampled a 10-dimensional representation with 2 clusters per class and 3 informative features. The more complex representation has 16 features, 3 clusters per class, and 3 informative features. The results (see https://anonymous.4open.science/r/RelativeFlatnessAndGeneralization-B175/AdditionalExperiments/genBoundApproxHardData.pdf) are again similar. The additional experiments indicate that the positive results do not depend on the architecture or data modality. We include these results in the final manuscript.

---

> > ### Author Response · Authors · 2021-08-12
> > **Suggested Related Work**
> >
> > We thank the reviewer for suggesting related work, we enjoyed reading the papers. We want to briefly discuss the relation to our work and will add a discussion on them to the related work section in the final manuscript.
> >
> > _On the distance between two neural networks and the stability of learning_, Bernstein et al., NeurIPS 2020:
> >
> > The paper analyzes when training a neural network converges to a minimum depending on its robustness to weight perturbations. Our paper instead analyzes how well such a trained model generalizes. It is an interesting question whether the approaches could be combined, e.g., by promoting feature robustness in each layer (e.g., by a suitable regularization term). For that, a combination of our analysis with deep relative trust seems promising.
> >
> > _The Goldilocks zone: Towards better understanding of neural network loss landscapes_, Fort et al., AAAI, 2019:
> >
> > The paper shows that standard initialization techniques initialize networks in a zone of high curvature - as measured by the fraction of positive Eigenvalues of the loss Hessian - and high Tr(H)/||H||. It is an interesting open question how high curvature at initialization - that leads to fast convergence - interacts with high flatness / low curvature at good minima.

---

> > > ### Comment · Reviewer_zbyb · 2021-08-31
> > > **A response to a response**
> > >
> > > I would like to thank the authors for a satisfactory and detailed reply to my comments. I am raising my score from 6 to 7 to reflect that some of the issues I pointed out have successfully been resolved. Thank you!

---

### Official Review · Reviewer_5qca · 2021-07-21

**Rating:** 6
**Confidence:** 4

**Summary:**

This paper describes a connection between flatness of minima and generalization in deep neural networks. The authors define a concept called feature robustness and show that it is related to flatness. This is derived through a straightforward observation that perturbations in feature space can be recast as perturbations of the model in parameter space. This allows the authors to define a flatness measure for minima in deep networks (this flatness measure is also invariant to rescalings of the layers in neural networks with positively homogenous activations). The authors combine their notion of feature robustness with epsilon representativeness of a function to connect flatness to generalization. They present empirical evaluations and comparison to other measures on CIFAR10.

**Limitations And Societal Impact:**

This is a theoretical paper that does not have immediate societal implications. The assumptions required for the analysis are outlined but a clearer discussion of the limitations of the conclusions is needed (this is only an upper bound on generalization, etc.)

**Main Review:**

This paper is able to once again confirm the relationship between flatness and generalization in an empirical manner. The measure presented in this paper shows better correlation to generalization than other measures proposed in the literature. However the authors do not provide the details about which layer was used to divide the model into the $\phi, \psi$ decomposition. The authors also do not specify whether the correlation holds for every possible choice of $\phi$ and $\psi$ to decompose the model. These details will help us understand the flatness phenomenon better.

The key theorem relating generalization and flatness is Theorem 6 (following from equation 5) which decomposes the generalization gap for a function $f$ into the $\epsilon$-representativeness of the feature mapping $\phi (S)$ for the distribution $\mathcal{D}$ and the feature robustness of the full model. The relationship between feature robustness and flatness follows directly from a Taylor expansion of the empirical loss, and the relationship between $\epsilon$-representativeness and generalization follows by definition. Proposition 2 is helpful in showing a relationship between local truncated normal distributions and orthogonal matrices. However, can one show a result of the form - for every local distribution $\lambda_i (\xi_i)$ there exists a distribution on linear perturbations $A$ such that the equivalence between feature perturbations and sampling from the distribution holds in expectation? It seems to me that this is an important step in establishing the link between feature robustness and generalization. Moreover, are there specific functions $\phi$ which make this possible, or does this hold for all feature mappings $\phi$? The synthetic experiments in Figures 3 and 4 are encouraging but are still performed with deep networks. Do the authors expect this to hold for deep linear networks, for instance?

Experiments on synthetic data showing the relationship between the feature robustness and $\epsilon$-representativeness terms would be helpful. What is the contribution of each term to the generalization gap? Is there a tradeoff between them or are more robust feature maps $\phi$ also more representative?

**Time Spent Reviewing:**

20

---

> ### Author Response · Authors · 2021-08-10
> **Response to Reviewer 5qca**
>
> We thank the reviewer for the considerable reviewing effort and valuable questions and suggestions. The reviewer appreciates the potential of the proposed measure and our approach to gain novel insight on the relation between flatness and generalization. In the following, we want to clarify the reviewer’s concerns, in particular about the decomposition into a feature representation and predictor function.
>
> **Which layer was used to divide the network?**
>
> Our theory applies to any model that can be decomposed as specified in line 40, i.e., it applies to any layer of a neural network and it also applies to deep linear networks (but it also applies, e.g., to manually crafted feature maps and linear models). We decided to provide experiments on networks that are often used in practice and applied the experiment on CIFAR10 on the feature representation of the last layer. However, the layer can be chosen freely. The main reason for our choice was runtime efficiency, since the last layer typically is less wide and thus the computation of the Hessian is less time consuming. We performed additional experiments comparing various layers of a neural network, but due to the limited time for the rebuttal, we performed them on  MNIST, where for each of the layers a strong correlation of flatness with the generalization can be observed. A plot of the results can be found in the anonymized git repository (https://anonymous.4open.science/r/RelativeFlatnessAndGeneralization-B175/AdditionalExperiments/MNIST_layerwise.png) with individual plots in the folder (https://anonymous.4open.science/r/RelativeFlatnessAndGeneralization-B175/AdditionalExperiments/MNIST_layerwise/). Note that these plots are generated with a simplified relative flatness measure that does not perform the renormalization necessary for neuron-wise reparameterization invariance. For the final version, we are running experiments on CIFAR10 with the relative flatness measure used throughout the paper.
>
>
> **Can Proposition 2 be strengthened to find a distribution A on matrices for each family of distributions $\lambda_i$?**
>
> This is a very interesting point. In general, it is not possible to find a distribution that precisely induces $\lambda_i$ for each $x_i$, since a single choice of linear perturbations of parameters affects each $x_i$, and the distribution on matrices cannot  be chosen for each sample point $x_i$ individually. We suspect that this nontriviality may be the main reason why the potential of our key equation (1) has been overlooked so far, and proving Proposition 2 is an achievement.
> However, it is possible to achieve 0-representativeness (equality in expectation) without inducing each $\lambda_i$ precisely.  There is a great flexibility in choosing the distribution on matrices. For sufficiently large delta, the mean value theorem can show the existence of a family of distributions with 0-representativeness. For given $\lambda_i$ and with a suitable distribution on matrices A, the corresponding flatness measure slightly changes from the trace to a (different) weighted sum over the eigenvalues of the Hessian. In other words, depending on the underlying distribution and the feature representation, a (situation-dependent) better flatness measure than the proposed one can exist, but is non-trivial to find. A comprehensive analysis of this would go beyond the scope of the paper, but we will add a discussion of this point in the final manuscript.
>
> **Is there a tradeoff between feature robustness and eps-representativeness?**
>
> There is indeed a trade-off between the two terms. For a delta-distribution on the zero-matrix, feature-robustness equals zero and the error term defining epsilon-representativeness equals the generalization gap (line 107). It requires nontrivial distributions on matrices to induce small error of representativeness. To derive a simple measure (using the trace), we choose the distributions of Proposition 2 and find that this simple, natural choice already outperforms competing measures (in particular after reparameterizations). We also point out that not arbitrary representations are considered when applying our measure, but to obtain a neuron-wise reparameterization independent network, only “balanced” representations are considered (see the normalization step in Theorem 4). There is a great potential to find more elaborate measures that correspond to more suitable distributions of matrices A which would lead to even tighter generalization bounds.
>
> Taken together, the representation- and also task-dependent derivation of more suitable distributions on matrices is non-trivial (and certainly a target for future work), but our simple and natural choice results in a natural measure of flatness that is shown to be already very performant, highlighting the potential of our approach to understand the relation between flatness and generalization. We explain these specificities in more detail in the final manuscript.

---

> > ### Comment · Reviewer_5qca · 2021-08-31
> > **Response to Authors**
> >
> > I thank the authors for addressing most of my questions. I appreciate the additional experiments that the authors have run on different layers. It seems to be the case that the metrics computed from all layers are correlated with generalization, which suggests that the correlation of flatness with generalization only requires a compositional model. It is here that my question about linear models or other types of compositional models is relevant, in my opinion.
> >
> > The authors answer to the relationship between $\lambda_i$ and $A_i$ is also a little confusing. If it is not possible to find an $A_i$ that induces $\lambda_i$ then it is unclear why multiplicative linear transformations are chosen to model the distributions $\lambda_i$. This could have been achieved in principle with a richer class of perturbation functions. The answer might be that the multiplicative linear transforms are related to flatness, but in that case the main result about the theoretical connection between flatness and generalization seems less convincing to me.
> >
> > I am also confused about the authors response to my question about whether there is a tradeoff between robustness and representativeness. I do not understand the point about the delta distribution on the zero matrix since that distribution is not $\epsilon$ robust for $\epsilon >0$. My question is about whether for a specific network high feature robustness implies low representativeness, or whether the two quantities are not linked in that manner.
> >
> > In light of these questions I am maintaining my rating.

---

> > > ### Author Response · Authors · 2021-09-01
> > > **Response to Reviewer**
> > >
> > > We thank the reviewer for the engaged discussion. Regarding the question about the trade-off between feature robustness and representativeness, we would like to emphasize that on the level of feature robustness, the error decomposition is an identity (equation in line 110 on page 3), not an upper bound. Therefore, the quantities are indeed linked and exactly characterize the generalization gap. Regarding the choice of multiplicative linear transforms, they were chosen to relate properties of the feature space to properties of the weight space (see Eq. 1). It is an interesting open question whether this can be done for richer classes of perturbation functions as well, but - as detailed in our initial reply - the chosen transforms already provide notable flexibility. It is unclear to us why this renders the connection between flatness and generalization less convincing. We would be grateful if the reviewer could elaborate this point.

---

### Author Response · Authors · 2021-08-25
**General Response to AC and Reviewers**

We thank the reviewers for their time and valuable feedback. All reviewers appreciate the capability of our theory to rigorously connect the local measure of flatness to generalization. The reviewers highlight that the paper “deals with an important question of the relationship between local loss Hessian based flatness and generalization” for which “there hasn’t been a satisfactory explanation for why this connection exists”. To this end, the paper shows “that generalization performance can be bounded by their flatness quantity”. “The [flatness] measure presented in this paper shows better correlation to generalization than other measures proposed in the literature” in an “empirical validation of their theoretical results using pretty large network[s] and a non-trivial dataset”. Furthermore, reviewers highlight that the “paper is pretty clearly written”, easy to follow and “contains both theory and an empirical evaluation." Most questions addressed the decomposition of the network function into a feature representation and a model, which we clarified in the individual answers to the reviewers. One question addressed the generality of the contribution, i.e., whether it only applies to networks where only the last layer is trainable. We clarified that the proposed theory applies to all models that can be expressed as a feature extractor and model as $f(x) =\psi(w,\phi(x))= g(w \phi(x))$, “which includes linear and kernel models, as well as most neural networks” and that the experiments have been carried out using standard training of ResNET 18. We have carefully studied all the review comments and addressed them in the manuscript.
We have not received feedback on our rebuttal. We would gladly clarify any remaining open questions.

---

### Decision · Program_Chairs · 2021-09-27

**Decision:**

Accept (Poster)

**Comment:**

This paper provides a theoretical account of the relation between flatness of minima and generalization in deep neural networks. This is an important problem that attracted recent attention. The results presented here (Theorem 6) provide novel insight and are validated by experiments. The presentation is clear.